# Remarkable catalytic activity of dinitrogen-bridged dimolybdenum complexes bearing NHC-based PCP-pincer ligands toward nitrogen fixation

Aya Eizawa[1], Kazuya Arashiba[1], Hiromasa Tanaka[2], Shogo Kuriyama[1], Yuki Matsuo[2], Kazunari Nakajima[1], Kazunari Yoshizawa[2,3] & Yoshiaki Nishibayashi[1]

Intensive efforts for the transformation of dinitrogen using transition metal–dinitrogen complexes as catalysts under mild reaction conditions have been made. However, limited systems have succeeded in the catalytic formation of ammonia. Here we show that newly designed and prepared dinitrogen-bridged dimolybdenum complexes bearing $N$-heterocyclic carbene- and phosphine-based PCP-pincer ligands [{Mo(N$_2$)$_2$(PCP)}$_2$($\mu$-N$_2$)] (**1**) work as so far the most effective catalysts towards the formation of ammonia from dinitrogen under ambient reaction conditions, where up to 230 equiv. of ammonia are produced based on the catalyst. DFT calculations on **1** reveal that the PCP-pincer ligand serves as not only a strong $\sigma$-donor but also a $\pi$-acceptor. These electronic properties are responsible for a solid connection between the molybdenum centre and the pincer ligand, leading to the enhanced catalytic activity for nitrogen fixation.

[1] Department of Systems Innovation, School of Engineering, The University of Tokyo, Bunkyo-ku, Tokyo 113-8656, Japan. [2] Institute for Materials Chemistry and Engineering, Kyushu University, Nishi-ku, Fukuoka 819-0395, Japan. [3] Elements Strategy Initiative for Catalysts and Batteries, Kyoto University, Nishikyo-ku, Kyoto 615-8520, Japan. Correspondence and requests for materials should be addressed to K.Y. (email: kazunari@ms.ifoc.kyushu-u.ac.jp) or to Y.N. (email: ynishiba@sys.t.u-tokyo.ac.jp).

Nitrogen is an essential element for all living things on earth. Since most of the nitrogen atoms on earth exist as the form of inert dinitrogen gas, the fixing of molecular dinitrogen is necessary to be utilized. The industrial dinitrogen fixation system, called the Haber–Bosch process, plays an important role in producing ammonia from dinitrogen gas today[1]. The operation of the process, however, requires high temperature and high pressure, resulting in large consumption of fossil fuels[1]. On the other hand, nitrogenases transform dinitrogen gas into ammonia under ambient reaction conditions, where the active sites of nitrogenases include iron, molybdenum and vanadium as essential transition metals[2–5]. Studies on the active sites of nitrogenases are considered to be important to elaborate an efficient artificial system for the catalytic ammonia formation from dinitrogen gas[6–8].

Despite intensive efforts for the transformation of dinitrogen gas using transition metal–dinitrogen complexes as catalysts under mild reaction conditions[9–16], only a few systems have succeeded in the catalytic formation of ammonia from dinitrogen gas[17–28]. In 2003, Yandulov and Schrock[29–32] reported the first successful example of the catalytic conversion of dinitrogen gas into ammonia using a molybdenum–dinitrogen complex as a catalyst and in 2013 Peters and co-workers[33–39] reported the iron-catalysed transformation using an iron–dinitrogen complex as a catalyst. We also found that several dinitrogen-bridged dimolybdenum complexes such as $[\{Mo(N_2)_2(PNP)\}_2(\mu-N_2)]$ (2; PNP = 2,6-bis(di-tert-butylphosphinomethyl)pyridine)[40–42] and molybdenum–nitride complexes bearing PNP-type pincer ligands[43] or mer-tridentate triphosphine[44] worked as more effective catalysts towards ammonia formation under ambient reaction conditions, where up to 63 equiv. of ammonia were produced based on the molybdenum atom of the catalyst.

During our continuous study, we have realized three promising clues to develop more effective catalysts. The first clue is the introduction of an electron-donating group to the pincer ligands to increase the backdonating ability of the molybdenum atom to the dinitrogen ligand. In fact, dinitrogen-bridged dimolybdenum complexes bearing the electron-donating group-substituted PNP-pincer ligands worked as more effective catalysts in our previous reaction system[41]. The second clue is the inhibition of the dissociation of the pincer ligand from the molybdenum atom to increase the stability of the molybdenum complex. We generally

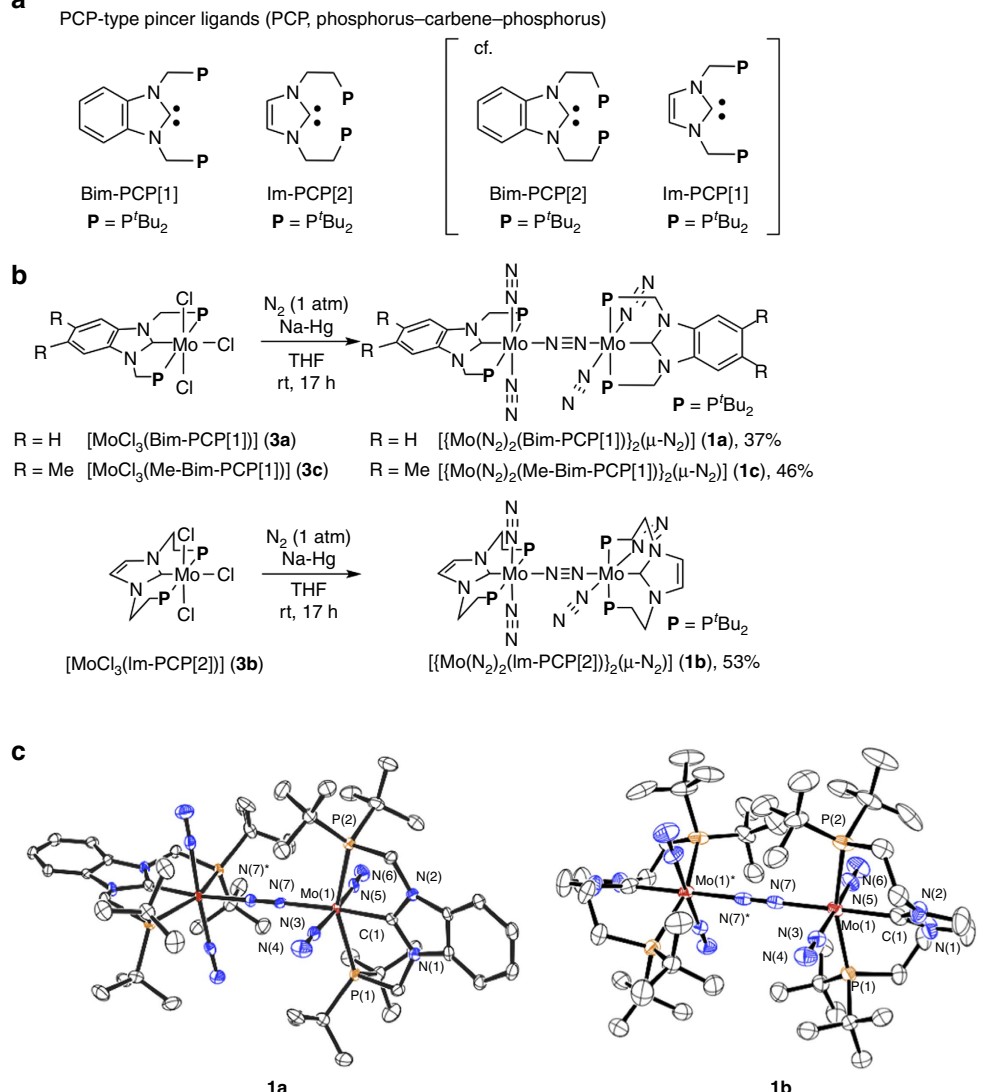

**Figure 1 | Design and synthesis of dinitrogen-bridged dimolybdenum complexes.** (**a**) Newly designed PCP-type pincer ligands (PCP, phosphorus–carbene–phosphorus). (**b**) Synthesis of dinitrogen-bridged dimolybdenum complexes **1a**–**1c**. (**c**) ORTEP drawings of **1a** (left) and **1b** (right). Thermal ellipsoids are shown at the 50% probability level. Hydrogen atoms and solvated molecules are omitted for clarity.

observed the dissociation of the PNP-pincer ligand after the ammonia formation in the catalytic reaction[40]. The third clue is the preservation of the dinitrogen-bridged dimolybdenum core to promote the catalytic ammonia formation from the coordinated dinitrogen[43]. Density functional theory (DFT) calculations demonstrated that the dinitrogen-bridged dimolybdenum structure plays a vital role in the protonation of a dinitrogen ligand, where one molybdenum moiety of the dinuclear molybdenum–dinitrogen complex works as a mobile ligand to the other molybdenum moiety as an active site[43].

Taking account of these clues, we have now planned to design an *N*-heterocyclic carbene- (NHC-)[45,46] and phosphine-based PCP-type pincer ligand (a PCP-type pincer ligand composed of NHC and two phosphines) as a tridentate ligand in place of the so far employed PNP-type pincer ligand for preparing a new molybdenum–dinitrogen complex. It is known that NHC works as a stronger electron-donating ligand than pyridine and binds to a transition metal centre more strongly than pyridine[47,48]. In this article, we demonstrate that dinitrogen-bridged dimolybdenum complexes bearing PCP-pincer ligands [{Mo(N$_2$)$_2$(PCP)}$_2$(μ-N$_2$)] (**1**) worked as effective catalysts towards ammonia formation under ambient reaction conditions, where up to 230 equiv. of ammonia were produced based on the catalyst (115 equiv. of ammonia based on the molybdenum atom of the catalyst). This is so far the most effective catalytic reduction of dinitrogen gas into ammonia under ambient reaction conditions using transition metal–dinitrogen complexes as catalysts.

## Results

**Preparation and characterization of 1.** On the basis of our proposal, we designed two types of dinitrogen-bridged dimolybdenum complexes bearing PCP-type pincer ligands with two *tert*-butyl groups on each phosphorus atom (Fig. 1a). One is the complex bearing methylene linkers between the NHC skeleton and the phosphorus atom, where a similar PCP-pincer ligand bearing two phenyl groups on each phosphorus atom has quite recently been reported by Rieger and co-workers[49]. The other is the complex bearing ethylene linkers, where similar PCP-pincer ligands bearing two phenyl groups on each phosphorus atom have already been reported by some research groups[50–55].

According to our previous procedure[40–42], we newly prepared three dinitrogen-bridged dimolybdenum complexes bearing the PCP-type pincer ligands. Treatment of [MoCl$_3$(PCP)] (**3a**, PCP = 1,3-bis((di-*tert*-butylphosphino)methyl)benzimidazol-2-ylidene (Bim-PCP[1]); **3b**, PCP = 1,3-bis(2-(di-*tert*-butylphosphino)ethyl)imidazol-2-ylidene (Im-PCP[2]); **3c**, PCP = 5,6-dimethyl-1,3-bis((di-*tert*-butylphosphino)methyl)benzimidazol-2-ylidene (Me-Bim-PCP[1])) with 6 equiv. of Na–Hg in tetrahydrofuran (THF) at room temperature for 17 h under an atmospheric pressure of molecular dinitrogen gave the corresponding dinitrogen-bridged dimolybdenum complexes [{Mo(N$_2$)$_2$(PCP)}$_2$(μ-N$_2$)] (**1a**; PCP = Bim-PCP[1], **1b**; PCP = Im-PCP[2], **1c**; PCP = Me-Bim-PCP[1]) in 37%, 53% and 46% yields, respectively (Fig. 1b). Detailed synthetic procedures for ligand precursors are included in Supplementary Methods and synthetic procedures for metal precursors **3a**–**3c** are included in Methods section. These dinitrogen-bridged dimolybdenum complexes were characterized by $^1$H and $^{31}$P{$^1$H} NMR. Detailed molecular structures of these complexes **1a**–**1c** were determined by X-ray crystallography (Fig 1c for **1a** and **1b**; Supplementary Fig. 1 for **1c**), which were similar to those of the dinitrogen-bridged dimolybdenum complexes bearing PNP-pincer ligands[40–42]. However, the bond lengths, the bond angles and dihedral angles were significantly different between **1a** and **1b** according to the nature of the linkers in the PCP-pincer ligands

(Supplementary Tables 3 and 4). The bond lengths defined by Mo(1)-C(1) of **1a** and **1b** were 2.064(2) Å and 2.153(4) Å, respectively, and the bond angles defined by P(1)-Mo(1)-P(2) of **1a** and **1b** were 153.95(3)° and 163.18(8)°, respectively. On the other hand, the dihedral angles defined by N(1)-C(1)-Mo(1)-N(5) of **1a** and **1b** were 82.35(5)° and 43.74(7)°, respectively. The shortened bond length of Mo(1)-C(1) of **1a** suggests the stronger π-backdonation from the molybdenum centre to the NHC unit due to the almost-perpendicular orientation of the NHC unit. On the other hand, the longer CH$_2$ linker of Im-PCP[2] forces the twisted coordination of the NHC and is likely to weaken the π-backdonation in **1b**. Further information on this topic is discussed based on DFT calculations (*vide infra*).

Infrared spectra of **1a**–**1c** in the solid state showed a strong absorption peak assignable to terminal dinitrogen ligands at 1,978, 1,911 and 1,969 cm$^{-1}$ respectively. The single peak of each complex corresponds to the dinitrogen-bridged dimolybdenum structure, as determined by X-ray crystallography. Compared with the infrared spectrum of **1b**, those of **1a** and **1c** showed the peak at much higher frequency due to the strong π-backdonation from the molybdenum centre to the NHC. The infrared spectra of **1a** and **1c** in THF solution showed one strong absorption peak assignable to terminal dinitrogen ligands at 1,979 and 1,973 cm$^{-1}$, respectively. Comparison of the infrared spectra of **1a** and **1c** in the solid state with that in the solution state indicates that the dinitrogen-bridged dinuclear structures of **1a** and **1c** are preserved even in solution. Furthermore, the $^{15}$N{$^1$H} NMR spectrum of $^{15}$N$_2$-labelled **1a** in C$_6$D$_6$ under $^{15}$N$_2$ showed two singlet and one doublet signals; δ 7.2 (s, Mo–*N*≡*N*–Mo), −13.0 (d, $^1J_{N-N}$ = 5.4 Hz, Mo–*N*≡*N*), −32.0 (br s, Mo–*N*≡*N*), which are consistent with the dinuclear structure (Fig. 2a)[40,56]. In contrast, the infrared spectrum of **1b** in THF solution showed two peaks assignable to terminal dinitrogen ligands, suggesting that the structure of **1b** in THF is no longer the same with that in the solid state. To obtain more information on real species of **1b** in the THF solution, the $^{15}$N{$^1$H} NMR spectrum of **1b** was measured in THF-$d_8$ solution under an atmospheric pressure of $^{15}$N$_2$ gas. The spectrum showed two doublet and two double triplet signals; δ 1.2 (d, $^1J_{N-N}$ = 6.0 Hz, Mo–*N*≡*N*(equatorial)), −20.8 (d, $^1J_{N-N}$ = 6.1 Hz, Mo–*N*≡*N*(axial)), −27.8 (dt, $^1J_{N-N}$ = 6.0 Hz and $^2J_{N-P}$ = 1.6 Hz, Mo–*N*≡*N* (equatorial)), −30.5 (dt, $^1J_{N-N}$ = 6.1 Hz and $^2J_{N-P}$ = 2.3 Hz, Mo–*N*≡*N*(axial); Fig. 2b), demonstrating the formation of the corresponding mononuclear dinitrogen complex [Mo(N$_2$)$_3$(Im-PCP[2])] (**1b′**) in THF. In fact, these spectroscopic features of **1b′** in THF are consistent with those of similar mononuclear dinitrogen complexes such as *mer*-[Mo(N$_2$)$_3$L$_3$] structures[57,58]. The instability of **1b** in solution may be derived from the steric repulsion between the two molybdenum moieties bearing Im-PCP[2].

As shown in Fig. 1, we prepared two types of the PCP-pincer ligands, where Bim-PCP[1] has a benzimidazol-2-ylidene skeleton and Im-PCP[2] has an imidazol-2-ylidene skeleton. Unfortunately, we were unable to synthesize the corresponding two PCP-pincer ligands based on the same NHC skeleton bearing linkers of the different lengths between the NHC skeleton and each phosphorus atom (Bim-PCP[2] and Im-PCP[1] in Fig. 1a). However, we consider that the presence or absence of benzene ring has little influence on either the thermodynamic stability of the Mo–N≡N–Mo structures or the electron-donating ability of the pincer ligands for the following reasons. To assess the influence of the benzene moiety in Bim-PCP[1] on the thermodynamic stability of the dinitrogen-bridged dimolybdenum structure, we calculated a model complex [{Mo(N$_2$)$_2$(Im-PCP[1])}$_2$(μ-N$_2$)], where the benzimidazol-2-ylidene skeleton in Bim-PCP[1] is replaced by the imidazol-2-ylidene skeleton. The optimized distance of the Mo–N$_2$(bridging) bond and its bond

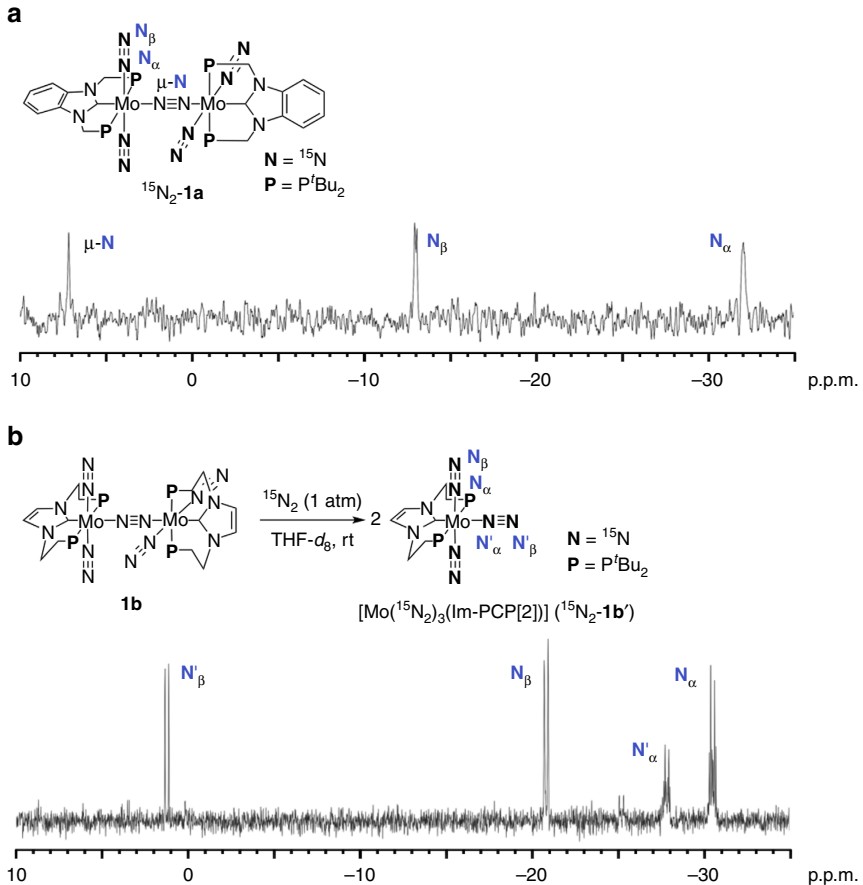

**Figure 2 | $^{15}N\{^1H\}$ NMR spectra of 1a and 1b.** (**a**) $^{15}N\{^1H\}$ NMR spectrum of $^{15}N_2$-**1a** in $C_6D_6$ under $^{15}N_2$. (**b**) $^{15}N\{^1H\}$ NMR spectrum of $^{15}N_2$-**1b'** in THF-$d_8$ under $^{15}N_2$.

dissociation energy (BDE) are 2.105 Å and 18.4 kcal mol$^{-1}$, respectively, both of which are almost identical to those calculated for **1a** (2.108 Å and 18.8 kcal mol$^{-1}$). The definition of BDE is described in the Methods section. On the other hand, Tuczek and co-workers[52,53] have previously prepared two PCP ligands based on benzimidazol-2-ylidene and imidazol-2-ylidene, where these ligands have similar σ-donating ability. Gusev[59,60] has previously estimated the donor ability of various NHC ligands based on the computational evaluation of $v_{CO}$ (A1) of [Ni(CO)$_3$(NHC)]. The author showed that 1,3-dimethylbenzimidazol-2-ylidene and 1,3-dimethylimidazol-2-ylidene have almost the same values of $v_{CO}$ (2,057 and 2,054 cm$^{-1}$, respectively), suggesting that the electron-donating ability of the NHC ligands was scarcely influenced by the difference between benzimidazol-2-ylidene and imidazol-2-ylidene. We therefore expect that the introduction of a benzene ring to the NHC skeleton little influences either the thermodynamic stability of the Mo–N≡N–Mo structure or the electron-donating ability of the pincer ligand.

We then performed DFT calculations to elucidate how the length of linkers connecting the NHC skeleton with the P$^t$Bu$_2$ groups in Bim-PCP[1] and Im-PCP[2] influences the thermodynamic stability of the dinitrogen-bridged dimolybdenum structures in **1a** and **1b**. Figure 3 shows optimized structures of dimolybdenum complexes **1a** and **1b**. The N≡N stretching frequencies of terminal dinitrogen ligands calculated for **1a** (2,012 cm$^{-1}$) and **1b** (1,969 cm$^{-1}$) reproduced the experimental trend. The Mo–N and N–N distances of a terminal dinitrogen ligand are calculated to be 2.032 and 1.137 Å for **1a** and 2.016 and 1.142 Å for **1b**, respectively, indicating that the terminal dinitrogen ligand in **1b** is more activated than that in **1a**. As a

result, the BDE of a Mo–N$_2$(terminal) of **1a** (11.9 kcal mol$^{-1}$) is considerably lower than that of **1b** (16.7 kcal mol$^{-1}$). On the other hand, the Mo–N distance of the bridging dinitrogen ligand of **1a** (2.108 Å) is shorter than that of **1b** (2.133 Å). The BDE of the Mo–N$_2$(bridging) bond of **1a** is 18.8 kcal mol$^{-1}$, which is more than twice as high as that of **1b**, 9.0 kcal mol$^{-1}$. The very low BDE of the Mo–N$_2$(bridging) bond of **1b** can be associated with the experimental observation that the dinuclear complex **1b** is labile to be separated into two mononuclear complexes in solution.

Differences in thermodynamic stability of the Mo–N≡N–Mo structure between **1a** and **1b** can be rationalized by optimized structures of the corresponding mononuclear dinitrogen complexes. Figure 4 presents space-filling models of [Mo(N$_2$)$_3$(PCP)] (**1a'**; PCP = Bim-PCP[1], **1b'**; PCP = Im-PCP[2]). Comparison of the Mo–N distance for the equatorial dinitrogen ligand of **1a'** (2.084 Å) with that of **1b'** (2.041 Å) suggests that **1b'** bearing Im-PCP[2] strongly binds dinitrogen at the equatorial position. Contrary to the BDEs of the Mo–N$_2$(bridging) bond calculated for dinuclear complexes **1a** and **1b**, the BDE of the Mo–N$_2$(equatorial) bond of **1b'** (21.5 kcal mol$^{-1}$) is almost the same with that of **1a'** (21.2 kcal mol$^{-1}$). The dramatic decrease in the Mo–N$_2$(equatorial) BDE of **1b'** in the formation of the dinuclear structure can be ascribed to steric hindrance caused by the *tert*-butyl groups on each phosphorus atom in Im-PCP[2]. The optimized structure of **1a'** bearing Bim-PCP[1] with the methylene linkers has the P–Mo–P bond angle of 151.2°, while that of **1b'** bearing Im-PCP[2] with the ethylene linkers has the bond angle of 164.3°. The extension of the CH$_2$ linkers in **1b'** forces the *tert*-butyl groups on each phosphorus atom in

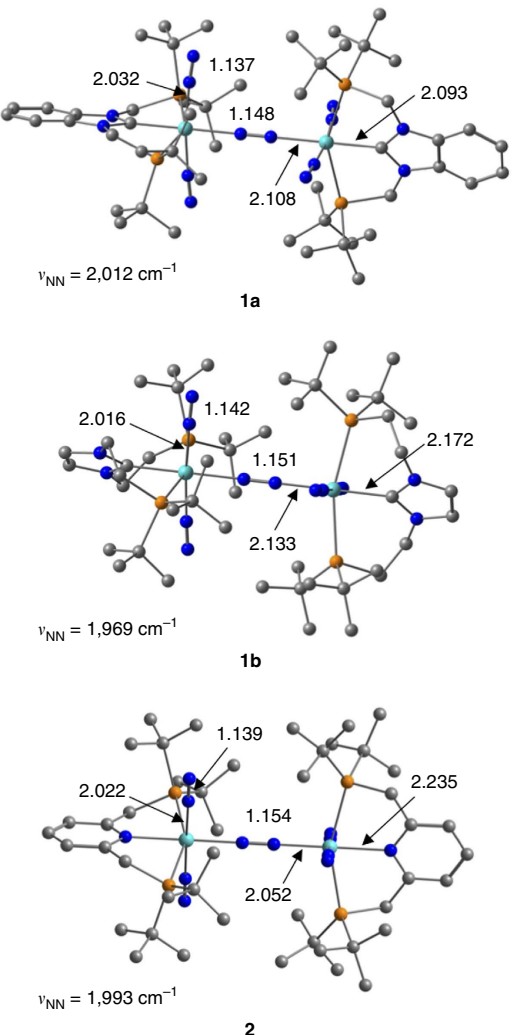

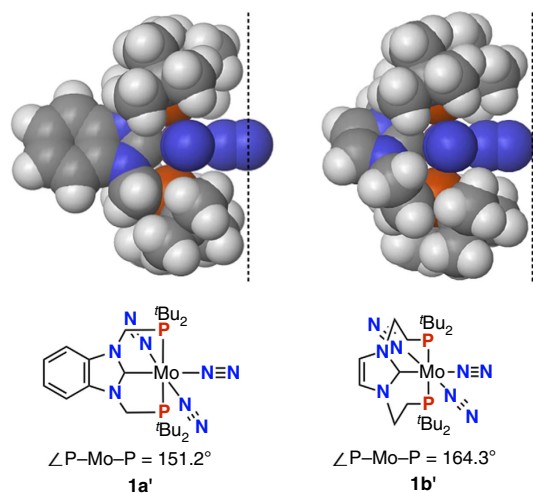

**Figure 4 | Space-filling models of mononuclear Mo–N$_2$ complexes 1a′ and 1b′.** The dashed lines represent the projection of *tert*-butyl groups on phosphine atoms.

**Figure 3 | Optimized structures of dinuclear complexes.** Bond distances are shown in Å. The values of $\nu_{NN}$ present the N≡N stretching frequencies of terminal dinitrogen ligands. Hydrogen atoms are omitted for clarity.

Im-PCP[2] to project towards the space surrounding the equatorial dinitrogen ligand (Fig. 4). As a result, the formation of a thermodynamically stable dimolybdenum complex bearing Im-PCP[2] is encumbered by steric repulsions between *tert*-butyl groups in two mononuclear molybdenum units facing each other.

**Catalytic nitrogen fixation using 1 as catalysts**. The catalytic reduction of molecular dinitrogen into ammonia using **1** as catalysts was carried out according to the following procedure of the previous method[40–42]. To a mixture of **1** and 2,6-lutidinium trifluoromethanesulfonate (96 equiv. to **1**; [LutH]OTf) as a proton source in toluene was added a solution of metallocene (72 equiv. to **1**) as a reductant in toluene via a syringe pump at room temperature over a period of 1 h, followed by stirring at room temperature for another 19 h under an atmospheric pressure of dinitrogen. After the reaction, the amounts of ammonia and molecular dihydrogen were determined by indophenol method[61] and gas chromatography (GC), respectively. The yields of ammonia and molecular dihydrogen were calculated based on the metallocene. Typical results are shown in Table 1. In all cases, no formation of other products such as hydrazine was observed at all.

First, we carried out the catalytic reaction in the presence of **1a** as a catalyst using either cobaltocene (CoCp$_2$; Cp = η$^5$–C$_5$H$_5$), decamethylchromocene (CrCp*$_2$; Cp* = η$^5$–C$_5$Me$_5$), and decamethylcobaltocene (CoCp*$_2$) as reductants, to produce 5.7, 17.6 and 11.8 equiv. of ammonia based on the catalyst, respectively (Table 1, runs 1–3). In the absence of a reductant, only 0.2 equiv. of ammonia were produced based on **1a** (Table 1, run 4). We have previously obtained the result that 12.2 equiv. of ammonia were produced based on the catalyst from the reaction with CrCp*$_2$ as a reductant in the presence of [{Mo(N$_2$)$_2$(PNP)}$_2$ (μ-N$_2$)] **2** as a catalyst (Table 1, run 9)[40]. This means that **1a** promoted the catalytic nitrogen fixation more effectively than **2**. Separately, we confirmed the direct conversion of molecular dinitrogen into ammonia by using $^{15}$N$_2$ gas in place of normal $^{14}$N$_2$ gas (see Supplementary Methods for detailed procedure).

In stark contrast to the catalytic activity of **1a**, **1b** did not work so effectively towards the formation of ammonia under the same reaction conditions. When CoCp$_2$, CrCp*$_2$ and CoCp*$_2$ were used as reductants, only 1.4, 3.2 and 2.9 equiv. of ammonia were produced based on the catalyst, respectively (Table 1, runs 5–7). In the absence of a reductant, 1.5 equiv. of ammonia were produced based on **1b** (Table 1, run 8).

Next, we investigated the influence of a proton source in the catalytic nitrogen fixation using **1a** as a catalyst. Typical results are shown in Table 2, where larger amounts of both reductant and proton source were employed in order to sharpen the difference among the results (see Supplementary Methods for the detailed procedure). The catalytic reaction using larger amounts of reductant CrCp*$_2$ (360 equiv. to **1a**) and proton source [LutH]OTf (480 equiv. to **1a**) afforded 79 equiv. of ammonia based on the catalyst (Table 2, run 1). When 2-picolinium trifluoromethanesulfonate ([PicH]OTf; Pic = 2-picoline) was used in place of [LutH]OTf, only a small amount of ammonia was produced based on the catalyst (Table 2, run 2). On the other hand, 2,4,6-collidinium trifluoromethanesulfonate ([ColH]OTf; Col = 2,4,6-collidine) worked rather effectively, where 61 equiv. of ammonia were produced (Table 2, run 3). When a non-coordinating anion BAr$^F_4{}^-$ (Ar$^F$ = 3,5-(CF$_3$)$_2$C$_6$H$_3$) was used in place of OTf$^-$ in [LutH]OTf, only a small amount of ammonia was produced (Table 2, run 4). These results indicate that the use of [LutH]OTf as a proton source is an essential factor to achieve the high performance of **1a** as a catalyst.

**Table 1 | Catalytic formation of ammonia from dinitrogen gas employing 1a or 1b as a catalyst.**

$$N_2 + 6 \text{ reductant} + 6 \text{ [LutH]OTf} \xrightarrow[\text{toluene, rt, 20 h}]{\text{cat. (10.0 μmol)}} 2 NH_3$$

1 atm     72 equiv.*     96 equiv.*

| Run | Catalyst | Reductant | $E_{1/2}$ (V)[†] | NH$_3$ (equiv.)* | NH$_3$ (%)[‡] | H$_2$ (equiv.)* | H$_2$ (%)[‡] |
|---|---|---|---|---|---|---|---|
| 1 | 1a | CoCp$_2$ | −1.15 | 5.7 ± 0.6 | 24 | 15.2 ± 1.5 | 42 |
| 2 | 1a | CrCp*$_2$ | −1.35 | 17.6 ± 0.7 | 73 | 3.9 ± 0.4 | 11 |
| 3 | 1a | CoCp*$_2$ | −1.78 | 11.8 ± 1.0 | 49 | 3.3 ± 0.8 | 9 |
| 4 | 1a | none | — | 0.2 | 0.8 | 0.1 | 0.3 |
| 5 | 1b | CoCp$_2$ | −1.15 | 1.4 ± 0.1 | 6 | 16.7 ± 1.9 | 47 |
| 6 | 1b | CrCp*$_2$ | −1.35 | 3.2 ± 0.3 | 13 | 15.1 ± 0.7 | 42 |
| 7 | 1b | CoCp*$_2$ | −1.78 | 2.9 ± 1.0 | 12 | 3.0 ± 1.0 | 8 |
| 8 | 1b | none | — | 1.5 | 6 | 0.1 | 0.3 |
| 9[§] | 2 | CrCp*$_2$ | −1.35 | 12.2 | 51 | 4.2 | 12 |

A solution of a reductant in 4 ml of toluene was added to a mixture of the catalyst and [LutH]OTf in 1 ml of toluene at room temperature over a period of 1 h, followed by stirring at room temperature for another 19 h under 1 atm of dinitrogen gas.
*Equivs based on catalyst
[†]$E_{1/2}$ values of reductant in MeCN versus Ag/Ag$^+$ (ref. 69).
[‡]Yields based on reductant.
[§]ref. 40.

**Table 2 | Catalytic formation of ammonia from dinitrogen gas using 1a as a catalyst.**

$$N_2 + 6 \text{ CrCp*}_2 + 6 \text{ proton source} \xrightarrow[\text{toluene, rt, 20 h}]{\text{cat. 1a (2.0 μmol)}} 2 NH_3$$

1 atm     360 equiv.*     480 equiv.*

| Run | Proton source | p$K_a$[†] | NH$_3$ (equiv.)* | NH$_3$ (%)[‡] | H$_2$ (equiv.)* | H$_2$ (%)[‡] |
|---|---|---|---|---|---|---|
| 1 | [LutH]OTf | 6.77 | 79 ± 4 | 66 | 12 ± 2 | 6 |
| 2 | [PicH]OTf | 5.97 | 19 ± 1 | 16 | 58 ± 11 | 28 |
| 3 | [ColH]OTf | 7.48 | 61 ± 1 | 51 | 28 ± 8 | 12 |
| 4 | [LutH]BAr$^F_4$ | 6.77 | 15 ± 5 | 13 | 42 ± 15 | 16 |

A solution of CrCp*$_2$ in 4 ml of toluene was added to a mixture of 1a and a proton source in 1 ml of toluene at room temperature over a period of 1 h, followed by stirring at room temperature for another 19 h under 1 atm of dinitrogen gas.
*Equivs based on catalyst.
[†]p$K_a$ value of proton source in H$_2$O (ref. 70).
[‡]Yield based on CrCp*$_2$.

$$N_2 + 6 \text{ CrCp*}_2 + \text{[LutH]OTf} \xrightarrow[\text{toluene, rt, 20 h}]{\text{cat. (1.0 μmol)}} 2 NH_3 + H_2$$

1 atm     1,440 equiv.     1,920 equiv.

cat.

**1a**
P = P$^t$Bu$_2$
NH$_3$ (2.0±0.2)×10$^2$ equiv. (42%)
H$_2$ (1.0±0.4)×10$^2$ equiv. (14%)

**1c**
P = P$^t$Bu$_2$
NH$_3$ 2.3×10$^2$ equiv. (48%)
H$_2$ 1.2×10$^2$ equiv. (16%)

**Figure 5 | Catalytic formation of ammonia using larger amounts of a reductant and a proton source in the presence of 1a or 1c as a catalyst.** A solution of CrCp*$_2$ in 5 ml of toluene was added to a mixture of the catalyst and [LutH]OTf in 1 ml of toluene at room temperature over a period of 1 h (for **1a**) or 5 h (for **1c**), followed by stirring at room temperature for another 19 h (for **1a**) or 15 h (for **1c**) under 1 atm of dinitrogen gas. The amounts of ammonia and hydrogen (equiv.) are based on the catalyst. Yields are based on CrCp*$_2$.

On the basis of the results shown in Tables 1 and 2, we carried out the catalytic reaction using much larger amounts of CrCp*$_2$ and [LutH]OTf as a reductant and a proton source to the catalyst, respectively (see Supplementary Methods for the detailed procedure). The reaction using 1,440 equiv. of CrCp*$_2$ as a reductant and 1,920 equiv. of [LutH]OTf as a proton source in the presence of **1a** as a catalyst under ambient reaction conditions gave $2.0 \times 10^2$ equiv. of ammonia based on the catalyst (Fig. 5). The catalytic activity of **1a** towards the formation of ammonia is an order of magnitude greater than that of [{Mo(N$_2$)$_2$(PNP)}$_2$ ($\mu$-N$_2$)] **2** (up to 23 equiv. of ammonia based on **2**). Ammonia was obtained in 42% yield together with molecular dihydrogen (14% yield). Furthermore, a higher catalytic activity has been achieved when **1c** was used as a catalyst, where up to $2.3 \times 10^2$ equiv. of ammonia based on the catalyst were produced under similar reaction conditions (Fig. 5). We have not yet obtained the exact reason why **1c** worked as the more effective catalyst than **1a**, but we consider that the introduction of two methyl groups to the benzimidazol-2-ylidene skeleton in the PCP-pincer ligand may increase the backdonating ability of molybdenum centres to the coordinated dinitrogen ligand and activated the terminal dinitrogen ligands more strongly than **1a**. The infrared spectrum of **1c** in the solid state showed a strong absorption peak assignable to terminal dinitrogen ligands at 1,969 cm$^{-1}$, which is lower than that of **1a** (*vide supra*). Previously, we reported that the introduction of electron-donating groups such as methyl and methoxy groups to the pyridine ring of the PNP-pincer ligand in **2** markedly enhanced the catalytic activity under the same reaction conditions[41].

The time profile of the catalytic reactions using **1a** and **1c** as catalysts was monitored (see Supplementary Methods for the detailed procedure). Typical results are shown in Fig. 6 together with the time profile using **2** as a catalyst[41,42]. The turnover frequency (TOF) for ammonia formation, which was determined as mols of ammonia (based on the catalyst) produced in initial 1 h, was 42 h$^{-1}$ for **1a** and 53 h$^{-1}$ for **1c**. The TOFs for ammonia using **1a** and **1c** are ca. 2.5 and 3.1 times larger than that using **2** (17 h$^{-1}$), respectively. This result indicates that the dinitrogen-bridged dimolybdenum complexes bearing PCP[1]-type pincer ligands such as **1a** and **1c** have the effective performance not only on the catalytic activity but also on the rate for ammonia formation.

For comparison of the stability of the dinitrogen-bridged dimolybdenum complexes bearing PCP-pincer ligands with that of PNP-pincer ligands, we carried out the following catalytic reactions using **1a** and **2** as catalysts. After the formation of 47 equiv. of ammonia based on the catalyst from molecular dinitrogen following the same procedure of the time profile experiment using **1a**, the same amounts of [LutH]OTf and CrCp*$_2$ were further added at room temperature, and the mixture was stirred for another 2 h to afford a total of 69 equiv. of ammonia based on the catalyst (Fig. 7; see Supplementary Methods for the detailed procedure). In this reaction system, 22 equiv. of ammonia were produced from a further reaction of molecular dinitrogen with excess amounts of a proton source and a reductant. This experimental result indicates that the active species derived from **1a** remain even after the catalytic reaction. In fact, no free PCP-pincer ligand was observed from the reaction mixture after the catalytic reaction using **1a** as a catalyst, suggesting that the active species derived from **1a** were still active. In sharp contrast, no additional ammonia was produced from similar treatment using **2** as a catalyst (Fig. 7), where free PNP-pincer ligand was observed from the reaction mixture after the catalytic reaction using **2** as a catalyst[40]. These results indicate that the stability of **1a** is much improved compared with that of **2**.

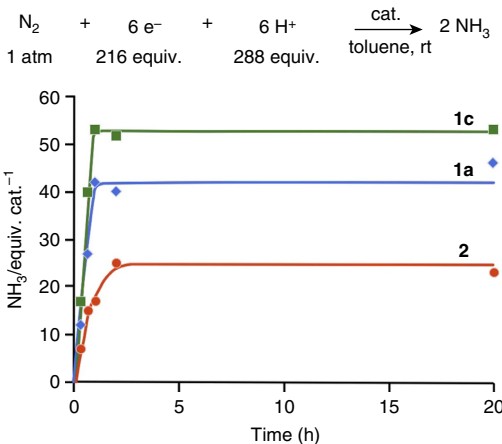

**Figure 6 | Time profiles of the formation of ammonia from dinitrogen gas.** A solution of CrCp*$_2$ (0.72 mmol) in toluene was added to a mixture of **1a** or **1c** (0.0033 mmol) and [LutH]OTf (0.96 mmol) at room temperature over a period of 1 h under 1 atm of dinitrogen gas, followed by stirring for the indicated time (0.33, 0.67, 1, 2 and 20 h). A solution of CoCp$_2$ (2.16 mmol) in toluene was added to a mixture of **2** (0.010 mmol) and [LutH]OTf (2.88 mmol) at room temperature over a period of 1 h under 1 atm of dinitrogen gas, followed by stirring for the indicated time (0.33, 0.67, 1, 2 and 20 h). The amount of ammonia (equiv.) is based on the catalyst.

**Comparison of PCP and PNP ligands.** In this section, we compare the electronic properties and reactivity of **1a** and **2**. We have previously reported that the catalytic activity of **2** was improved by the introduction of electron-donating groups to the 4-position of the pyridine ring in PNP[41]. In this report, DFT calculations demonstrated that the introduction of electron-donating groups to PNP enhances the backdonating ability of molybdenum centres and thereby leads to activation of dinitrogen ligands. As described in the Introduction, the NHC-based PCP ligand was expected to work as a strong electron donor to activate dinitrogen ligands coordinated to the molybdenum centre. For understanding the geometric and electronic structures of **1a** and **2** upon the coordination of the pincer ligands, mononuclear molybdenum complexes **1a'** and **2'** were investigated in detail.

Figure 8a compares the electron-donating ability of Bim-PCP[1] and PNP in terms of differences in atomic charge ($\Delta q$) between the dinitrogen complexes (**1a'** and **2'**) and the free ligands (Bim-PCP[1] and PNP). The charges of the Mo atom and three N$_2$ ligands obtained with the natural population analysis (NPA)[62] were set to zero for the free ligands, and hence the sum of the charges is identical to the $\Delta q$ value of the Mo(N$_2$)$_3$ moiety. The gross NPA charges of the Mo(N$_2$)$_3$ moiety in **1a'** and **2'** can be regarded as the amount of electron donated from the pincer ligands during complexation. To evaluate the electron-donating ability of Bim-PCP[1] and PNP, the $\Delta q$ values of the P$^t$Bu$_2$ groups and the carbene or pyridine moiety containing the methylene linkers are separately given in Fig. 8a. The $\Delta q$ values of the P$^t$Bu$_2$ groups are identical in both Bim-PCP[1] and PNP (+0.29), indicating that the electron-donating ability of Bim-PCP[1] and PNP is controlled by the carbene and pyridine moieties. Since the $\Delta q$ values of the carbene and pyridine moieties are +0.23 and +0.12, respectively, the pincer ligands donate 0.81e$^-$ (Bim-PNP[1]) and 0.70e$^-$ (PNP) to the Mo(N$_2$)$_3$ moiety during complexation. As we expected, the NHC-based pincer ligand exhibits a stronger electron-donating ability than the pyridine-based one from a viewpoint of atomic charge.

Optimized structures and BDEs between the molybdenum centre and dinitrogen ligands also reflect the strength of the

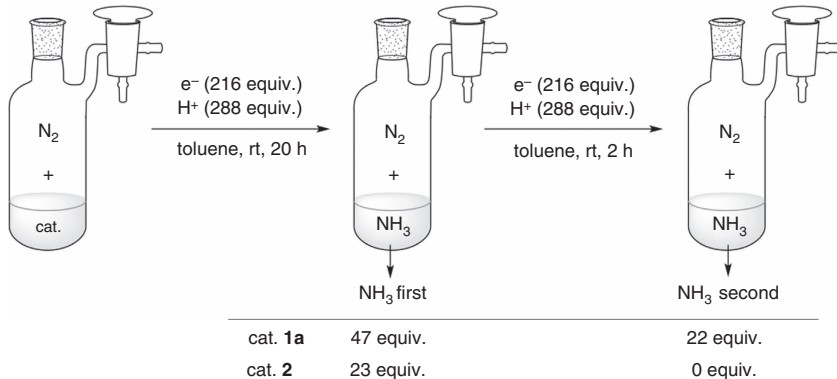

**Figure 7 | Reactions of further addition of proton source and reductant.** 'NH$_3$ first' and 'NH$_3$ second' were collected in separated runs. Each 'NH$_3$ first' is the same value as the time profile experiment. Each 'NH$_3$ second' is collected by the following procedure. A solution of a reductant (CrCp*$_2$ for **1a** and CoCp$_2$ for **2**; 216 equiv) in toluene (4 ml) was added to a mixture of **1a** or **2** (0.0033 mmol for **1a** and 0.010 mmol for **2**) and [LutH]OTf (288 equiv) at room temperature over a period of 1 h under 1 atm of dinitrogen gas, followed by stirring for 20 h. Then, [LutH]OTf (288 equiv) was added in one portion and another solution of the same reductant (216 equiv) in toluene (4 ml) was added over a period of 1 h, followed by stirring at room temperature for another 1 h under 1 atm of dinitrogen gas. The difference between the amount of ammonia obtained in this experiment and the 'NH$_3$ first' is the 'NH$_3$ second'. The amount of ammonia (equiv.) is based on the catalyst.

electron-donating ability of Bim-PCP[1] and PNP. Table 3 summarizes geometric parameters around the molybdenum centre in mononuclear molybdenum–dinitrogen complexes **1a**′ and **2**′, together with the BDEs of the Mo–N$_2$(axial) and Mo–N$_2$(equatorial) bonds. The Mo-C(carbene) bond distance (2.099 Å) in **1a**′ is significantly shorter than the Mo–N(pyridine) distance (2.240 Å) in **2**′. The Mayer bond order (b.o.)[63] of the Mo-C(carbene) bond is calculated to be 0.91, which is much larger than that of the Mo–N(pyridine) bond (0.39). The presence of a strong bonding interaction between the molybdenum centre and Bim-PCP[1] is consistent with the experimental fact that **1a** bearing Bim-PCP[1] works as a long-lived catalyst for the catalytic nitrogen fixation with a high turnover number compared with **2** bearing PNP (*vide supra*). On the other hand, the strong trans influence of the carbene ligand weakens the Mo–N$_2$(equatorial) bond in **1a**′. The Mo–N$_2$(equatorial) bond distance of 2.084 Å (b.o. = 0.50) in **1a**′ is much longer than that of 2.018 Å (b.o. = 0.62) in **2**′, and the BDEs of the Mo–N$_2$(equatorial) bond are 21.2 kcal mol$^{-1}$ for **1a**′ and 30.1 kcal mol$^{-1}$ for **2**′. Interestingly, the coordination of Bim-PCP[1] to molybdenum would influence all dinitrogen ligands at both *trans*- and *cis*-positions so as to weaken all the Mo–N$_2$ bonds. The Mo–N$_2$(axial) bond distances (b.o.) are calculated to be 2.034 Å (0.53) for **1a**′ and 2.024 Å (0.54) for **2**′. The BDE of the Mo–N$_2$(axial) bond of **1a**′ (12.5 kcal mol$^{-1}$) is also lower than that of **2**′ (14.0 kcal mol$^{-1}$). A similar trend was observed for the Mo–N$_2$(bridging) and Mo–N$_2$(terminal) bonds in dimolybdenum–dinitrogen complexes **1a** and **2**. The bond dissociation energies are 18.8 and 11.9 kcal mol$^{-1}$ for **1a**, both of which are smaller than those obtained for **2** (24.9 and 14.4 kcal mol$^{-1}$).

The origin of the weaker Mo–N$_2$ bonds in **1a**′ and **1a** is understood by looking at frontier orbitals responsible for the bonding between the molybdenum centre and the carbene C atom of Bim-PCP[1][52,53,64,65]. As depicted in Fig. 8b, the HOMO-6 (**1a**′) and HOMO-5 (**2**′) contribute to a σ-bond between the Mo atom and the carbene C atom (or the pyridine N atom). The large size of the lobe between the Mo and C atoms indicates that Bim-PCP[1] works as a strong σ-donor compared to PNP. The HOMO-1 in Fig. 8c mainly contributes to π-backdonation from an out-of-plane *d* orbital of Mo to a π* orbital of dinitrogen ligands. The backdonation from metal to dinitrogen is essential for the activation of dinitrogen upon the

formation of metal–dinitrogen complexes. Occupation of the HOMO-1 strengthens all of the Mo–N$_2$ bonds in **1a**′ and **2**′ because of their symmetrical structures. By comparing the HOMO-1 of **1a**′ and **2**′, one can find a bonding interaction between the Mo atom and the carbene C atom through π-backdonation from the *d* orbital of Mo to the vacant *p* orbital of C perpendicular to the carbene ring in **1a**′. This backdonation decreases the amount of electron transferred to both the equatorial and axial dinitrogen ligands, leading to the lower BDEs of the Mo–N$_2$ bonds in **1a**′ (**1a**). As presented in Fig. 8a, the Δ$q$ value of the dinitrogen ligands in **1a**′ (−0.26) is smaller than that in **2**′ (−0.31) in spite of the electron-donating ability of Bim-PCP[1] superior to PNP. The backdonation from the Mo atom to the carbene C atom also contributes to the strong binding of Bim-PCP[1] to Mo.

On the other hand, the Mo-C bond distance (b.o.) in **1b**′ bearing Im-PCP[2] (2.178 Å (0.79)) indicates that the Mo-C bond in **1b**′ is weaker than that in **1a**′, as summarized in Table 3. Owing to the longer CH$_2$ linkers, the coordination of the carbene moiety to the molybdenum centre in **1b**′ is highly twisted compared to **1a**′; the dihedral angles of N(1)-C(1)-Mo(1)-N(3) are 69.6° for **1a**′ and 43.8° for **1b**′ (Supplementary Fig. 21). The twisted coordination of the carbene moiety in **1b**′ reduces the overlap between the out-of-plane *d* orbital of the Mo atom and the vacant *p* orbital of the carbene C atom. As a result, Im-PCP[2] works only as a very strong σ-donor (0.90e$^-$ donation to Mo; Fig. 8a). The gross NPA charge on the dinitrogen ligands (−0.36) as well as the large BDE of the Mo–N$_2$(axial) bond of **1b**′ (14.3 kcal mol$^{-1}$) implies that the coordination of Im-PCP[2] to the molybdenum centre effectively activates the coordinated dinitrogen ligands. However, we theoretically confirmed that the mononuclear molybdenum–dinitrogen complexes such as **1b**′ and [*cis*-Mo(N$_2$)$_2$(Im-PCP[2])] cannot be protonated by LutH$^+$, similar to **1a**′ and [Mo(N$_2$)$_3$(PNP)][43]. All attempts to optimize a product complex comprise the protonated **1a**′ (**1b**′), and Lut resulted in formation of a reactant complex comprising **1a**′ (**1b**′) and LutH$^+$, even though the optimization started from a structure with the N$_2$…H$^+$ distance of 5 Å. Thus, the lack of the catalytic activity of **1b** for nitrogen fixation can be attributed to the thermodynamic instability of the dinitrogen-bridged dimolybdenum structure, as mentioned in the former section.

On the basis of the catalytic reaction pathway previously proposed for nitrogen fixation using **2** (ref. 43), we have

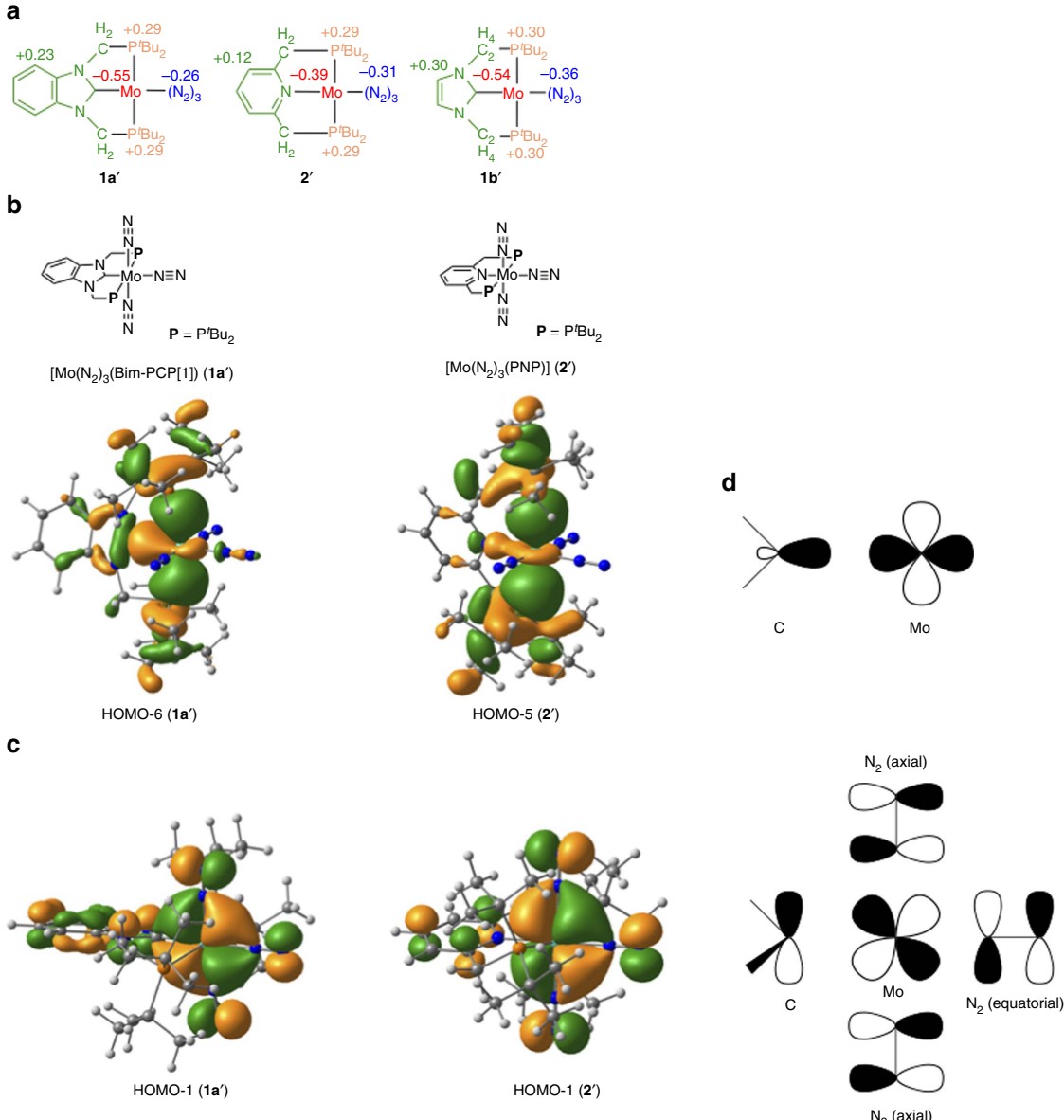

**Figure 8 | Electronic properties of mononuclear molybdenum–dinitrogen complexes.** (**a**) Changes in the NPA atomic charge ($\Delta q$) in the coordination of the pincer ligands to the Mo($N_2$)$_3$ moiety. The values of $\Delta q$ are obtained as differences between mononuclear Mo–$N_2$ complexes (**1a′**, **2′** and **1b′**) and free ligands (Bim-PCP[1] for **1a′**, PNP for **2′** and Im-PCP[2] for **1b′**). (**b**) Spatial distribution of frontier orbitals of **1a′** and **2′** that contribute to $\sigma$ donation from the pincer ligand to Mo. (**c**) Spatial distribution of frontier orbitals of **1a′** and **2′** that contribute to $\pi$ back donation from Mo to both equatorial and axial dinitrogen ligands. The molecular structures are rotated by 90° along the Mo–$N_2$(equatorial) bond from those in Fig. 8b. (**d**) A schematic drawing of the bonding interactions between the Mo atom and the carbene C atom in **1a′**.

theoretically investigated possible reaction pathways catalysed by **1a**. In the present article, we particularly focus on the first protonation process shown in Fig. 9, since the protonation of a terminal dinitrogen ligand in **2** by [LutH]OTf is energetically the most unfavourable process in the catalytic cycle[43]. In the calculated reaction pathway, a terminal dinitrogen ligand in **1a** is first protonated by LutH$^+$ (**1a**→**A**-PCP), and then the dinitrogen ligand *trans* to the generated NNH group is eliminated (**A**-PCP→**B**-PCP). The protonation of **1a** yielding **A**-PCP is endothermic by 8.1 kcal mol$^{-1}$ with an activation energy of 8.3 kcal mol$^{-1}$. This energy profile indicates that proton detachment from **A**-PCP can easily occur like the PNP system. On the other hand, the following $N_2$ elimination yielding **B**-PCP is exothermic by 5.2 kcal mol$^{-1}$ with a low activation energy of 4.0 kcal mol$^{-1}$. The coordination of OTf$^-$ to **B**-PCP is highly

exothermic by 20.7 kcal mol$^{-1}$, and thus the whole reaction pathway leading to **C**-PCP is energetically downhill. Comparison of the energy profiles of the PCP and PNP systems suggests that the reactivity of the dinitrogen complexes **1a** and **2** with [LutH]OTf would not be a major factor for rationalizing the high catalytic activity of **1a**.

## Discussion

On the basis of our previous findings on the catalytic nitrogen fixation, we have newly designed and prepared novel dinitrogen-bridged dimolybdenum complexes bearing NHC and phosphine-based PCP-pincer ligands, Bim-PCP[1] and Im-PCP[2]. The dimolybdenum–dinitrogen complexes bearing Bim-PCP[1] as PCP-pincer ligands have been found to work as so far the most effective catalysts towards the ammonia formation from

**Table 3 | Selected bond distances in Å and Mo–N₂ BDEs in kcal mol⁻¹ of mononuclear molybdenum–dinitrogen complexes.**

|  | 1a′ | 2′ | 1b′ |
|---|---|---|---|
| Mo–C(carbene)/N(pyridine) | 2.099 (0.91) | 2.240 (0.39) | 2.178 (0.79) |
| Mo–N(equatorial) | 2.084 (0.50) | 2.018 (0.62) | 2.041 (0.55) |
| Mo–N(axial) | 2.034 (0.53) | 2.024 (0.54) | 2.026 (0.53) |
| BDE(equatorial $N_2$) | 21.2 | 30.1 | 21.5 |
| BDE(axial $N_2$) | 12.5 | 14.0 | 14.3 |

BDE, bond dissociation energy.
The Mayer bond orders are presented in parenthesis.

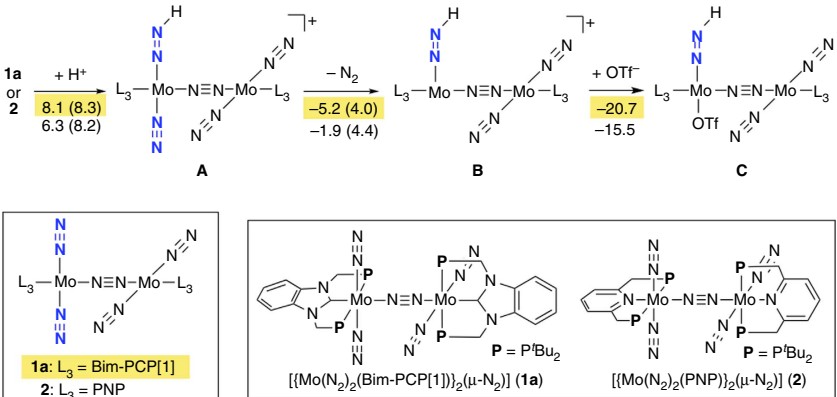

**Figure 9 | A possible reaction pathway and energy profiles of the first protonation process on a terminal dinitrogen ligand in 1a (highlighted in yellow) and 2.** Energy changes (activation energies in parentheses) are presented in kcal mol⁻¹.

molecular dinitrogen under ambient reaction conditions, where up to 230 equiv. of ammonia were produced based on the catalyst (115 equiv. of ammonia based on the molybdenum atom of the catalyst). The superior activity of dimolybdenum–dinitrogen complexes bearing Bim-PCP[1] included the high TOF for ammonia formation and the catalyst stability. DFT calculations on 1 reveal that Bim-PCP[1] as a PCP-pincer ligand serves as not only a strong σ-donor but also a π-acceptor. These electronic properties are responsible for a solid connection between the molybdenum centre and the pincer ligand, leading to the enhanced catalytic activity for nitrogen fixation.

## Methods

**General information.** Detailed experimental procedures, characterization of compounds and the computational details can be found in the Supplementary Figs 1–21, Supplementary Tables 1–10 and Supplementary Methods. Cartesian coordinates are available in Supplementary Data 1.

**Synthesis of [MoCl₃(PCP)] (3a–3c).** A typical procedure for the preparation of **3a** is described below. To a mixed solid of 1,3-bis((di-*tert*-butylphosphino)methyl)-1*H*-benzo[*d*]imidazol-3-ium hexafluorophosphate (**7a**, 1.16 g, 2.00 mmol) and KN(SiMe₃)₂ (559 mg, 2.80 mmol) was added toluene (40 ml), and the resulting suspension was stirred for 20 min at room temperature. [MoCl₃(thf)₃] (733 mg, 1.75 mmol) and toluene (15 ml) were added to the suspension and stirred at 80 °C for 19 h. The solvent was removed under vacuum, and the residue was washed with hexane (5 ml × 2), toluene (10 ml) and hexane (5 ml × 2). The solid was dried under vacuum. The solid was extracted with CH₂Cl₂ (10 ml × 1, 5 ml × 7), recrystallized from CH₂Cl₂–hexane and dried under vacuum to afford **3a** · 0.5CH₂Cl₂ (712 mg, 1.05 mmol, 60%). Anal. Calcd. for C₂₅.₅H₄₅Cl₄MoN₂P₂

(**3a** · 0.5CH₂Cl₂): C, 45.08; H, 6.68; N, 4.12. Found: C, 45.41; H, 6.61; N, 4.44. Crystals suitable for preliminary X-ray analysis were prepared by recrystallizing from CH₂Cl₂–hexane to give **3a** · CH₂Cl₂ as orange-brown crystals.

**3b:** Recrystallization from CH₂Cl₂–hexane gave **3b** · 0.5CH₂Cl₂ as orange crystals. 33% yield. Anal. Calcd. for C₂₃.₅H₄₇Cl₄MoN₂P₂ (**3b** · 0.5CH₂Cl₂): C, 42.94; H, 7.21; N, 4.26. Found: C, 43.04; H, 7.44; N, 4.19. Crystals suitable for X-ray analysis were prepared by recrystallizing from 1,2-dichloroethane–hexane to afford **3b** · 1/3C₆H₁₄. The structure is included in Supplementary Fig. 2 and selected bond lengths and angles in **3b** are included in Supplementary Table 6.

**3c:** Recrystallization from CH₂Cl₂–hexane afforded **3c** · 0.5CH₂Cl₂ as orange crystals. 48% yield. Anal. Calcd. for C₂₇.₅H₄₉Cl₄MoN₂P₂ (**3c** · 0.5CH₂Cl₂): C, 46.69; H, 6.98; N, 3.96. Found: C, 46.58; H, 6.79; N, 4.07. The structure is included in Supplementary Fig. 3, and selected bond lengths and angles in **3c** are included in Supplementary Table 7.

**Synthesis of [{Mo(N₂)₂(PCP)}₂(μ-N₂)] (1a–1c).** A typical procedure for the preparation of **1a** is described below. To a suspension of Na–Hg (0.5 wt% Na, 13.8 g, 3.00 mmol) in THF was added [MoCl₃(Bim-PCP[1])] · 0.5CH₂Cl₂ (341 mg, 0.501 mmol), and the resulting suspension was stirred under atmospheric pressure of N₂ at room temperature for 17 h. The supernatant suspension was filtered through Celite, and the solvent was removed under vacuum. The resulting solid was extracted with benzene (5 ml) and filtered through Celite. The filter cake was washed with benzene (2 ml × 9), and the solvent of the combined solution was removed under vacuum. The resulting solid was washed with pentane (2 ml × 20) to give **1a** · 1.3C₄H₈O · 0.4C₅H₁₂ as a dark purple solid, where 1.3 equiv. of THF and 0.4 equiv. of hexane were determined by ¹H NMR (141 mg, 0.102 mmol, 37%). Analytically pure sample was prepared by recrystallization from THF at −18 °C to afford **1a**.

**1a:** ¹H NMR (C₆D₆): δ 7.01–6.98 (m, 4H, Ar*H*), 6.75–6.71 (m, 4H, Ar*H*), 3.84 (s, 8H, NC*H*₂), 1.38 (pseudo t, ³$J_{P-H}$ = 5.7 Hz, 72H, P*ᵗBu*₂). ³¹P{¹H} NMR (C₆D₆): 105.6 (s, *PᵗBu*₂). Infrared (KBr, cm⁻¹): 1,978 (s, $v_{NN}$). Infrared (THF, cm⁻¹): 1,979 (s, $v_{NN}$). Anal. Calcd. for C₅₀H₈₈Mo₂N₁₄P₄: C, 50.00; H, 7.38; N, 16.33. Found: C, 49.66; H, 6.94; N, 14.18. The lower content of nitrogen is

considered to be due to the labile property of the coordinated dinitrogen ligand in **1a** under the analytical conditions.

**1b**: Recrystallization from benzene–hexane afforded **1b** · $2/3C_6H_{14}$ as dark-brown crystals. 53% yield. Anal. Calcd. for $C_{50}H_{101.33}Mo_2N_{14}P_4$ (**1b** · $2/3C_6H_{14}$): C, 49.44; H, 8.41; N, 16.14. Found: C, 49.82; H, 8.40; N, 15.35. $^1H$ and $^{31}P\{^1H\}$ NMR were measured in THF-$d_8$ as a mixture of **1b** and **1b′**. $^1H$ NMR (THF-$d_8$): **1b**, $\delta$ 6.76 (s, 2H, NC*H*C*H*N), 1.44 (pseudo t, $^3J_{P-H}=11.2$ Hz, P$^tBu_2$), 0.95 (pseudo t, $^3J_{P-H}=11.2$ Hz, P$^tBu_2$): **1b′**, $\delta$ 6.85 (s, 2H, NC*H*C*H*N), 4.29–4.19 (br m, 4H, NC*H*$_2$), 1.98–1.91 (br m, 4H, C*H*$_2$P), 1.18 (br s, 36H, P$^tBu_2$). $^{31}P\{^1H\}$ NMR (THF-$d_8$): **1b**, $\delta$ 71.1 (s, $^tBu_2$P): **1b′**, $\delta$ 69.2 (s, $^tBu_2$P). Infrared (KBr, cm$^{-1}$): 1,911 (s, $\nu_{NN}$ for **1b**). Infrared (THF under $N_2$, cm$^{-1}$): 2,041 (m, $\nu_{NN}$ for **1b′**), 1,945 (s, $\nu_{NN}$ for **1b**).

**1c**: Reprecipitation from THF–hexane afforded **1c** · $0.5C_6H_{14}$ as a brown solid. 46% yield. Crystals suitable for X-ray analysis were prepared by recrystallization from THF at $-18$ °C to afford **1c**. The structure is included in Supplementary Fig. 1, and selected bond lengths and angles are included in Supplementary Table 5. $^1H$ NMR ($C_6D_6$): $\delta$ 6.68 (s, 4H, Ar*H*), 3.89 (s, 8H, NC*H*$_2$P), 2.26 (s, 12H, ArC*H*$_3$), 1.41 (pseudo t, $^3J_{P-H}=5.5$ Hz, 72H, P$^tBu_2$). $^{31}P\{^1H\}$ NMR ($C_6D_6$): $\delta$ 105.8 (s, $P^tBu_2$). Infrared (KBr, cm$^{-1}$): 1,969 (s, $\nu_{NN}$). Infrared (THF, cm$^{-1}$): 1,973 (s, $\nu_{NN}$). Anal. Calcd. for $C_{57}H_{103}Mo_2N_{14}P_4$ (**1c** · $0.5C_6H_{14}$): C, 52.65; H, 7.98; N, 15.08. Found: C, 52.80; H, 7.70; N, 13.37.

**Catalytic reduction of dinitrogen to ammonia under $N_2$.** In a 50 ml Schlenk flask were placed **1a** · $1.3C_4H_8O$ · $0.4C_5H_{12}$ (12.9 mg, 0.00970 mmol) and 2,6-lutidinium trifluoromethanesulfonate [LutH]OTf (247 mg, 0.960 mmol). Toluene (1.0 ml) was added under $N_2$ (1 atm), and then a solution of CrCp*$_2$ (232 mg, 0.719 mmol) in toluene (4.0 ml) was added to the stirred suspension in the Schlenk flask with a syringe pump at a rate of 4.0 ml h$^{-1}$. After the addition of CrCp*$_2$, the mixture was further stirred at room temperature for 19 h. The reaction mixture was evaporated under reduced pressure, and the distillate was trapped in dilute $H_2SO_4$ solution (0.5 M, 10.00 ml). Aqueous solution of potassium hydroxide (30 wt%; 5 ml) was added to the residue to fully liberate ammonia, and the mixture was distilled into another dilute $H_2SO_4$ solution (0.5 M, 10.00 ml). The amount of $NH_3$ present in each of the $H_2SO_4$ solutions was determined by the indophenol method.[61]

**Computational method.** DFT calculations were performed to search all intermediates and transition structures on potential energy surfaces using the Gaussian 09 programme.[66] Similar to the previous study[43], we adopted the B3LYP* functional, which is a reparametrized version of the B3LYP hybrid functional developed by Reiher et al.[67] For optimization, the Stuttgart–Dresden pseudopotentials and 6–31G(d) basis sets were chosen for the Mo atom and the other atoms, respectively. To determine the energy profile of the first protonation process, we performed single-point energy calculations at the optimized geometries using the 6–311 + G(d,p) basis sets in place of the 6–31G(d) basis sets. Zero-point energy corrections were applied for energy changes ($\Delta E$) and activation energies ($E_a$) calculated for each reaction step. Solvation effects (toluene) were taken into account by using the polarizable continuum model in the single-point energy calculations.[68] More details are described in Supplementary Methods. Throughout the paper, the BDE of an Mo–$N_2$ (terminal, axial or equatorial) bond is defined as the energy change for dissociation of the corresponding dative $N_2$ ligand, for example, $[\{Mo(N_2)_2(PNP)\}_2(\mu\text{-}N_2)] \rightarrow [\{Mo(N_2)(PNP)\}\text{-NN-}\{Mo(N_2)_2(PNP)\}] + N_2$. The BDE of the Mo–$N_2$ (bridging) bond is defined as the energy change for separation of a dimolybdenum complex into two mononuclear Mo–$N_2$ complexes, such as $[\{Mo(N_2)_2(PNP)\}_2(\mu\text{-}N_2)] \rightarrow [Mo(N_2)_3(PNP)] + cis\text{-}[Mo(N_2)_2(PNP)]$.

**Data availability.** The X-ray crystallographic coordinates for structures reported in this article have been deposited at the Cambridge Crystallographic Data Centre (CCDC), under deposition number CCDC 1482254 (**1a**), 1482255 (**1b**), 1482256 (**1c**), 1482258 (**3b**) and 1482259 (**3c**). These data can be obtained free of charge from The Cambridge Crystallographic Data Centre via www.ccdc.cam.ac.uk/data_request/cif. All other data are available from the authors upon reasonable request.

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

## Acknowledgements

The present project is supported by CREST, JST. We thank KAKENHI (Nos JP26288044, JP26105708, JP15K13687, JP15H05798 to Y.N., No. JP24109014 to K.Y. and No. JP26888008 to H.T.) from JSPS and MEXT. A.E. and S.K. are recipients of the JSPS Predoctoral Fellowships for Young Scientists. We also thank the Research Hub for Advanced Nano Characterization at The University of Tokyo for X-ray analysis.

## Author contributions

K.Y. and Y.N. directed and conceived this project. A.E., K.A., S.K. and K.N. conducted the experimental work. H.T. and Y.M. conducted the computational work. All authors discussed the results and wrote the manuscript.

## Additional information

**Competing interests:** The authors declare no competing financial interests.

