## [Peer review file · Nature Communications]

Reviewers' comments:

Reviewer #1 (Remarks to the Author):

This submission by Nishibayashi and coworkers describes a new family of N-heterocyclic carbene "PCP" pincer ligated molybdenum complexes that exhibit exceptional activity for catalytic reduction of dinitrogen to ammonia using protons and electrons. The manuscript can be generally considered as having two major parts. The first section details the synthesis, characterization and catalytic activity of the new molybdenum compounds while the second part of the manuscript serves to rationalize the observed reactivity trends with mechanistic studies and computational (DFT) analysis of frontier orbitals and bonding. Overall the paper seems exceedingly long for publication in a "Communications" journal.

The first part of the manuscript detailing the synthesis and characterization of the new PCP-based molybdenum dinitrogen compounds is very thorough. The authors characterize the title complexes rigorously both in solution as well as in the solid state. The synthesis of the $\text{PCPMo}(\text{N}_2)_3$ compound is a strong addition to the coordination chemistry literature and a remarkable compound in its own right. A detailed investigation of the catalytic activities of the complexes was conducted and 1 was found to produce equivalents (230 eqs. per Mo) of ammonia under conditions employing a large excess of strong acid ($[\text{LutH}][\text{OTf}]$) and strong reductant (CrCp^*2). These results validate the authors' design principle for the application of PCP ligand system, whereby a more electron donating chelates yield higher numbers of turnovers.

A key experiment that should be conducted in recording the IR spectrum of the dinitrogen complexes collected in the presence of 96 equivalents of $[\text{LutH}][\text{OTf}]$. A color change in this compound and the previously reported PNP compounds upon addition of acid. Do the N_2 ligands survive under these conditions? If not the mechanistic hypotheses are not very useful!

In the second section of the manuscript, the authors aim to provide a rationale for the high catalytic activity of the newly synthesized PCP pincer system. While potentially very instructive, there are some issues with this section that need clarification and overall improvement. First and foremost, the authors examine the first hypothetical protonation event in complex 1, a step that has been previously demonstrated to be the thermodynamically most difficult step in a hypothetical catalytic cycle in the previously reported bimetallic PNP systems. This seems to be a reasonable assumption. The protonation event consists of three elementary steps proceeding through two possible pathways, as depicted in Figure 8; either protonation of a terminal N_2 ligand followed by N_2 dissociation trans to the site of protonation and subsequent $[\text{OTf}]^-$ coordination (Path I), or N_2 dissociation followed by protonation and $[\text{OTf}]^-$ ligation (Path II). The authors compute the relative thermodynamic stabilities of the intermediates as well as the activation energies for these elementary steps and conclude that Path II is more favorable due to the "instability" of intermediate A, which has a low calculated barrier (0.1 kcal) to revert to 1a. Why is path II more favored? Pathway I has a significantly lower overall activation barrier (12.3 kcal/mol vs. 20.8 kcal/mol in Path II) and should be favored, despite the low activation barrier for the reversion of A to 1a. By the authors' argument, if the forward reaction from intermediate A is inaccessible under Curtin-Hammett conditions due to a relatively low barrier for the reverse reaction, then intermediate B is far less likely to be formed in the first place given that its formation has an associated activation barrier nearly 8 kcal/mol higher than the overall barrier of path I. From the computational data presented, it is in fact the first protonation step in path I with an activation barrier of 8 kcal/mol that should be rate-determining, and not the N_2 dissociation of path II. The authors should clarify this point and all subsequent conclusions drawn from it.

Following the conclusion that N_2 dissociation is the critical step in a hypothetical catalytic cycle involving the bimetallic molybdenum PNP and PCP systems, the authors proceed to computationally

study the Mo-N₂ BDEs in the molybdenum dinitrogen complexes 1a and 2. The results of these calculations show exceedingly weak Mo-N₂ BDEs in both 1a and 2, with those in 1a being 3-6 kcal/mol weaker. The authors then show, through molecular orbital analysis, that the π -accepting nature of the PCP ligand framework serves as an "electron sink", decreasing the propensity of molybdenum for π -backbonding to the terminal N₂ ligands, thus resulting in weakened Mo-N bonds in the PCP systems. The authors then correlate this bond weakening with the higher catalytic activity with the PCP ligand framework, as per the earlier mechanistic proposal where the key step was determined to be N₂ dissociation. While potentially interesting, the authors must clarify how such Mo-N BDEs were calculated or define exactly what they mean. Use of the term BDE is confusing as typically this term refers to a homolytic process; translating this concept onto metal complexes where dissociation of a dative ligand renders clarification.

In summary, this work is interesting and could potentially shed light on some valuable design principles for synthesizing more effective catalysts for N₂ reduction. However, the authors must revisit and clarify their arguments, rationalization for the observed high catalytic activity of the newly synthesized PCP complexes and provide additional experimental support for their mechanistic hypotheses.

Specific comments:

1. The authors are correct in abbreviating their pincer ligand as PCP. However, that abbreviation has been used extensively in the organometallic literature by Goldman, Brookhart, Jensen, Kaska, Ozerov and many others to mean a pincer with a central phenyl donor. While not absolutely essential, the authors may want to consider another abbreviation for their ligand to avoid confusion. I frequently found myself reverting to the more classical definition during my reading of the manuscript.
2. Page 10: The authors showed the high stability of the newly synthesized PCP complexes, which producing more ammonia upon the addition of excess acid and reductant at the completion of the catalytic reaction whereas the PNP systems do not. Perhaps the higher performance of the NHC-based system is simply a result of a slower rate of decomposition (ligand dissociation) under the reaction conditions when compared with the original PNP design? This is potentially a valuable design principle worth explicitly mentioning.
3. Introduction is spelled incorrectly.
4. The connection between these compounds and nitrogenases are remote at best. While it is important to mention the enzymes in the introduction, the sentence "Mimicking the active sites..." seems misplaced. This paper doesn't strike me as biomimetic.

Reviewer #2 (Remarks to the Author):

Referee report on the manuscript entitled:

"Remarkable Catalytic Activity of Dinitrogen-Bridged Dimolybdenum-Dinitrogen Complexes Bearing NHC-Based PCP-Pincer Ligands toward Nitrogen Fixation"

By Nishibayashi et al.

The authors report on dinitrogen-bridged dimolybdenum-dinitrogen complexes containing N-heterocyclic carbene/phosphine ligands. These complexes are able to convert dinitrogen into ammonia under ambient conditions. Producing 230 equiv of ammonia, the new complex $[\text{Mo}(\text{N}_2)_2(\text{PCP})_2(\mu\text{-N}_2)]$ is the most effective catalyst known today. The reaction pathway and the electronic properties of the different ligands are investigated using DFT calculations.

Publication is recommended after consideration of the following points:

(1) IR spectroscopy has been performed on complexes 1a and 1b, leading to the observation of bands at 1978 cm^{-1} and 1911 cm^{-1} .

(i) These spectra should be shown.

(ii) These bands are assigned to the N-N stretching frequencies of the terminal N_2 ligands; i.e., they should correspond to the antisymmetric combination of N-N stretches at the individual Mo units.

However, it is to be anticipated that due to the coupling of the two Mo units these bands in turn split into two bands. For the first dinuclear penta(bisdinitrogen) complex of this type,

$[\{\text{W}(\text{N}_2)_2(\text{PEt}_2\text{Ph})_3\}_2(\mu\text{-N}_2)]$ such a splitting has in fact been observed (Anderson et al., J. Chem. Soc., Chem. Commun. 1982, 1291).

(iii) The symmetric combination of the N-N stretching frequencies of the terminal N_2 ligands should also appear as a small band in the IR spectrum. Moreover, these vibrations as well as the N-N stretch of the bridging ligand should be observable with Raman spectroscopy.

(iv) All of these frequencies should be discussed concerning the different ligand moieties.

(2) The ^{15}N NMR spectrum of 1b has been measured. It would be interesting to determine and discuss the coupling constants which are obtained, also in comparison to those of the dinuclear complex 1a.

(3) Regarding the issue of how the imidazole and benzimidazole based PCP ligands influence the properties of the resulting complexes the papers of Gradert et al. 2013 & 2014 should be cited, where exactly this point has been treated.

(4) There is no obvious reason why the second catalytic experiments (using CrCp^*2 and different proton sources) are performed in a different way than the first experiments. Why do the amounts of metallocene and proton sources are different in both experiments? To get comparable results it would be necessary to use the same equivalents of electron and proton source in both runs.

(5) By means of the catalytic experiments of the different benzimidazole based PCP ligands it becomes clear that the methylated ligand is the best catalyst. Can this observation be understood based on spectroscopy (i.e., IR spectroscopy and/or ^{15}N -NMR spectroscopy) coupled to DFT calculations?

(6) Pg. 13: Please document the observation of the formation of the dinitrogen-bridged dimolybdenum complex with loss of one terminal dinitrogen ligand by showing some data.

(7) ^{13}C -NMR experiments of some ligands and of the complexes 1a and 1b are missing. The measured ^{13}C -NMR spectra should be assigned. In particular, it is important to discuss the shift of the coordinating C-atom of the PCP ligands.

Recommendation: Publish after revision

Reviewer #3 (Remarks to the Author):

Summary:

This manuscript describes the design, synthesis, and characterization of an improved Mo-based catalyst for the formation of ammonia from molecular dinitrogen. It details mechanistic insights gained by altering the ligand on the catalyst. By changing the central donor atom from a phosphine to a more electron-donating carbene, there is a stronger interaction between the molybdenum center and the pincer ligand and the mechanism for ammonia formation also changes. The first step is loss of N₂ followed by protonation, which is distinct from the previously-reported catalyst. This ligand change enables a more efficient and effective catalyst for the formation of ammonia from molecular dinitrogen that suffers less from decreased catalytic over time via ligand dissociation from the catalyst. Furthermore, adding electron-donating substituents to the PCP ligand further improves the catalyst, presumably by increasing the π -backbonding ability of the molybdenum centers and increasing the activation of the dinitrogen ligands. It was also determined that the planar nature of the PCP ligand and Mo center allows for better orbital overlap, which decreases π -backbonding to the terminal dinitrogen ligands. The observations made in this manuscript shed light on the effects of changing a ligand's electronic and structural properties on catalyst activity, and this increased understanding could allow for the design of even more effective catalyst for nitrogen fixation. Thus this manuscript contains data and conclusions that would be interesting to readers of Nature Communications and should be acceptable for publication in this journal after some careful proofreading to streamline the English and consideration of the following points:

Comments:

- Some issues with prepositions throughout the paper (i.e., at the bottom of page 5: "...or the electron-donating ability of the pincer ligands for the following reasons," at the bottom of page 10 (and several times throughout the paper): "protonation of a terminal dinitrogen ligand," to/for mix-ups throughout, etc.)
- On page 7 at the end of the first paragraph, the reasoning for why the Mo-N₂(equatorial) bond distance in 1b' is significantly elongated when the dinuclear complex is formed is unclear to me. ("because the symmetrical dimolybdenum structure of 1 specifies the position of the bridging nitrogen at the midpoint of the two molybdenum atoms"...I am not sure why this would necessarily cause the Mo-N₂ to elongate significantly).
- Why is the activity of catalyst 1b the same in the presence of CoCp₂ and without a reductant present at all? Is this just a case of CoCp₂ not being a strong enough reducing agent for this compound?
 - Along the same lines, what are the redox potentials of 1a and 1b and why is CrCp*₂ so effective? I liked that the authors listed the potentials of the reductants but no conclusions were drawn from them.
- I think the term "dinitrogen-bridged dimolybdenum dinitrogen complexes" that is used throughout is unnecessarily wordy. Since you typically are also using a formula or specifying a compound number, you could probably just say "dinitrogen-bridged dimolybdenum complexes" and your readers would figure out that two axial N₂ ligands are also bound.
- The differences in Mo-C and N(1)-C(1)-Mo(1)-N5 angle are attributed to steric effects on page 4, but it becomes clear later in the paper that electronic effects are also important here too. I agree that the tBu groups stick out further when the ethylene linkers are used, but the orientation of the NHC ligand is also clearly very important electronically. Certainly Mo-C backbonding is the reason for the shorter Mo-C bond in 1a.
- Along the same lines, I think that the backbonding between Mo and the NHC ligand as a function of the orientation of the NHC needs to be brought up earlier in the manuscript. As soon as I saw the crystal structures, this became obvious to me, but it didn't actually get mentioned in the manuscript until the computational section. A reader should be able to understand these logical conclusions without computational data and orbital pictures. You can always refer the reader to the later portion of

the paper using "vide infra". I also think the authors should use the very different IR N2 stretching frequencies to bolster this point. I don't think the interplay between Mo-C backbonding and N2 stretching frequency was emphasized well in the computational section either – could have been correlated with BDE.

- Are any of the 3 peaks observed in the IR spectrum of 1b in solution attributed to remaining N2-bridged compound?
- What is the experimental error in the data points listed in Tables 1 and 2? Do each of these represent just one catalytic run? Typically, yields are reported as an average of 3 trials so that experimental error can be taken into account when comparing numbers.
- The experimental section is quite short and not reflective of all the experimental details in the paper. I think, at the very least, the synthesis of 1b should be reported here. I realize that much of the experimental detail has been relegated to the Supporting Information file, but I think the reader should be informed of that. For example, a sentence like "Detailed synthetic procedures for ligand precursors and metal precursors 3a-3c are included in the Supporting Information file." One specific catalytic run was described, but was this procedure also used for all of the catalytic runs in the paper? If so, this needs to be explicitly stated. If not, the modifications used for each trial need to be described (can be placed in the SI file, as long as the reader is told where to find it).
- Compound 1c seemingly comes out of nowhere when the catalysis is being discussed. Is this a new compound or was it previously reported elsewhere? If it is new, it should be included in the "preparation and characterization" section of the manuscript. If it is previously reported, this should be stated explicitly in the manuscript with a reference provided.
- In the "Accession codes" section of the manuscript, the CCDC numbers for 1c, and 3a-3c are listed. I didn't see these structures mentioned in the paper, and there is no compounds defined as 3c.
- Similarly, the "General methods" section of the Supporting information files mentions that 4a and 7b were prepared accordingly to literature methods, but these compounds aren't mentioned in the text. In fact, there are lots of new compound numbers defined in the SI file for ligand precursors. I think that a general scheme that shows how all the ligands were synthesized (and defines compound numbers ahead of time) would be useful for the reader before jumping right into synthetic methods.
- The functional and basis sets used for the DFT calculations should be mentioned in the main text of the paper (even in the "Methods" section would be fine). The reader can then be referred to the SI file for further details.
- These days, it is customary to include spectral data (1H NMR, 31P NMR) for all new compounds in the Supporting Information file, at least for diamagnetic compounds. This is particularly warranted in this case because proof of bulk purity (elemental analysis, high res mass spec) is not provided for any of the ligands or their precursors.
- Crystallography is generally convincing of the connectivity of the compounds, but there are some errors that need to be addressed or explained prior to publication. 1a- structure is OK. 1b- incompletely modelled solvent disorder. 1c-structure is OK. 3a –appears to be a poor quality structure without all atoms anisotropically refined. 3b - incompletely modelled solvent disorder. 3c – high value of Rint needs to be explained.
- There are many sections/paragraphs/sentences that could be reworded and/or shortened to be more clear and to the point. (i.e., on page 10: "After the formation of 47 equiv of ammonia...suggesting that the active species derived from 1a were still alive." This could all be reworded to be more clear and could be significantly condensed, as there is a lot of unnecessary/repetitive info being relayed). I would say "were still active" or "were still present", rather than "were still alive."
- Page 10, end of section – "These results indicate that the catalytic behavior of 1a is quite different from that of 2." I agree with this statement, but think that the authors could use more precise language rather than "catalytic behavior." Perhaps "catalyst activity and stability" would be more specific.
- Page 12, end of first paragraph states that "we also theoretically confirmed that six- and five coordinate mononuclear dinitrogen complexes, 1a' and B', can not be protonated by LutH+." How was this confirmed? Where is the data for this?

- Page 12, first two sentences of “Electronic properties of PCP[1] in 1a” section: should the references be 12d instead of 12e? 12d seems to be the correct citation and is also the citation used to cite the same fact on page 9.
- Page 12: parameters of the Mo-N₂(terminal) moiety are in Figure 3 (typo)
- Page 14: Why does the coordination of the more electron-donating Bim-PCP[1] ligand to molybdenum increase the Mo-Naxial distances as compared to those of the PNP complex?
- Page 15: How is the Δq of the dinitrogen ligands defined? It was previously defined as the difference in atomic charge between the mononuclear No-N₂ complexes and the free ligands (Bim-PCP[1] and PNP), but this doesn't seem applicable when describing the Δq of the dinitrogen ligands.
- Typos in references:
 - o 12e – page # should be 3940
 - o 13a – remove “No.”

Response to Referees' Comments

The followings are our answers to the comments.

As for the comments by Reviewer 1

(1) Reviewer 1 pointed out that "A key experiment that should be conducted in recording the IR spectrum of the dinitrogen complexes collected in the presence of 96 equivalents of [LutH]OTf. A color change in this compound and the previously reported PNP compounds upon addition of acid. Do the N₂ ligands survive under these conditions? If not the mechanistic hypotheses are not very useful!" According to the suggestion, we have newly performed the indicated protonation of PCP complex **1a** and PNP complex **2** with an excess amount of [LutH]OTf, respectively. A mixture of dinitrogen complex (**1a** or **2**) and 96 equivalents of [LutH]OTf in toluene was stirred at room temperature for five minutes. After the reaction, the colors of both reaction mixtures changed. The IR spectra of both supernatant solutions were recorded, however, the absorptions derived N₂ ligands were not observed due to the overlap with absorptions derived from [LutH]OTf. Further investigation to support our mechanistic hypothesis is currently in progress. Thank you very much for the valuable suggestions.

(2) Reviewer 1 pointed out that "In the second section of the manuscript, the authors aim to provide a rationale for the high catalytic activity of the newly synthesized PCP pincer system. While potentially very instructive, there are some issues with this section that need clarification and overall improvement. First and foremost, the authors examine the first hypothetical protonation event in complex **1**, a step that has been previously demonstrated to be the thermodynamically most difficult step in a hypothetical catalytic cycle in the previously reported bimetallic PNP systems. This seems to be a reasonable assumption. The protonation event consists of three elementary steps proceeding through two possible pathways, as depicted in Figure 8; either protonation of a terminal N₂ ligand followed by N₂ dissociation trans to the site of protonation and subsequent [OTf]⁻ coordination (Path I), or N₂ dissociation followed by protonation and [OTf]⁻ ligation (Path II). The authors compute the relative thermodynamic stabilities of the intermediates as well as the activation energies for these elementary steps and conclude that Path II is more favorable due to the "instability" of intermediate A, which has a low calculated barrier (0.1 kcal) to revert to **1a**. Why is path II more favored?"

Pathway I has a significantly lower overall activation barrier (12.3 kcal/mol vs. 20.8 kcal/mol in Path II) and should be favored, despite the low activation barrier for the reversion of A to 1a. By the authors' argument, if the forward reaction from intermediate A is inaccessible under Curtin-Hammett conditions due to a relatively low barrier for the reverse reaction, then intermediate B is far less likely to be formed in the first place given that its formation has an associated activation barrier nearly 8 kcal/mol higher than the overall barrier of path I. From the computational data presented, it is in fact the first protonation step in path I with an activation barrier of 8 kcal/mol that should be rate-determining, and not the N₂ dissociation of path II. The authors should clarify this point and all subsequent conclusions drawn from it". As pointed out by Reviewer 1, it would be difficult to conclude that Path II prefers to Path I in the PCP system because the large difference in activation energies for the protonation of N₂ and dissociation of N₂ should correspond with a large difference between their reaction rates. We still think that Path II is also a possible pathway for the first protonation because we observed that one of the dinitrogen ligands in **1a** was readily dissociated by drying a solid **1a** under vacuum. The corresponding sentences on pages 11 and 12 have been modified in the revised manuscript. We really appreciate your valuable comments.

- (3) Reviewer 1 pointed out that "Following the conclusion that N₂ dissociation is the critical step in a hypothetical catalytic cycle involving the bimetallic molybdenum PNP and PCP systems, the authors proceed to computationally study the Mo-N₂ BDEs in the molybdenum dinitrogen complexes 1a and 2. The results of these calculations show exceedingly weak Mo-N₂ BDEs in both 1a and 2, with those in 1a being 3-6 kcal/mol weaker. The authors then show, through molecular orbital analysis, that the π -accepting nature of the PCP ligand framework serves as an "electron sink", decreasing the propensity of molybdenum for π -backbonding to the terminal N₂ ligands, thus resulting in weakened Mo-N bonds in the PCP systems. The authors then correlate this bond weakening with the higher catalytic activity with the PCP ligand framework, as per the earlier mechanistic proposal where the key step was determined to be N₂ dissociation. While potentially interesting, the authors must clarify how such Mo-N BDEs were calculated or define exactly what they mean. Use of the term BDE is confusing as typically this term refers to a homolytic process; translating this concept onto metal complexes where dissociation of a dative ligand renders clarification". In the present paper, the BDEs of Mo-N₂ bonds are defined as follows: For terminal, axial, and equatorial N₂ ligands, the energy change for

dissociation of the corresponding dative N₂ ligand from Mo, for example, $[\text{Mo}(\text{N}_2)_2(\text{PNP})]_2(\mu\text{-N}_2) \rightarrow [\text{Mo}(\text{N}_2)(\text{PNP})\text{-NN-Mo}(\text{N}_2)_2(\text{PNP})] + \text{N}_2$. For the bridging N₂ ligand, the energy change for separation of a dimolybdenum complex into two mononuclear Mo-N₂ complexes, such as $[\text{Mo}(\text{N}_2)_2(\text{PNP})]_2(\mu\text{-N}_2) \rightarrow [\text{Mo}(\text{N}_2)_3(\text{PNP})] + \text{cis-}[\text{Mo}(\text{N}_2)_2(\text{PNP})]$. The definition of the BDEs has been added to the “Method” part.

- (4) Reviewer 1 pointed out that “Specific comments: 1. The authors are correct in abbreviating their pincer ligand as PCP. However, that abbreviation has been used extensively in the organometallic literature by Goldman, Brookhart, Jensen, Kaska, Ozerov and many others to mean a pincer with a central phenyl donor. While not absolutely essential, the authors may want to consider another abbreviation for their ligand to avoid confusion. I frequently found myself reverting to the more classical definition during my reading of the manuscript”. Thank you very much for your suggestion. However, some other research groups have already used PCP as the abbreviation for similar ligands. As a result, we have not changed the original abbreviation of PCP in the present manuscript.
- (5) Reviewer 1 pointed out that “Specific comments: 2. Page 10: The authors showed the high stability of the newly synthesized PCP complexes, which producing more ammonia upon the addition of excess acid and reductant at the completion of the catalytic reaction whereas the PNP systems do not. Perhaps the higher performance of the NHC-based system is simply a result of a slower rate of decomposition (ligand dissociation) under the reaction conditions when compared with the original PNP design? This is potentially a valuable design principle worth explicitly mentioning”. As pointed out by Reviewer 1, the experimental results shown in Figure 7 suggest the improvement in stability of PCP complex, not the reaction rate of PCP complex. We have modified the original description in the revised manuscript on page 11. Thank you for your valuable suggestions.
- (6) Reviewer 1 pointed out that “Specific comments: 3. Introduction is spelled incorrectly”. According to the suggestions, we have corrected the indicated spelling in the revised manuscript. Thank you very much for your careful reading.
- (7) Reviewer 1 pointed out that “Specific comments: 4. The connection between these compounds and nitrogenases are remote at best. While it is important to mention the enzymes in the introduction, the sentence “Mimicking the

active sites..." seems misplaced. This paper doesn't strike me as biomimetic". According to the suggestion, we have modified the original description in the revised manuscript. Thank you very much for your valuable suggestion.

As for the comments by Reviewer 2

(8) Reviewer 2 pointed out that "(1) IR spectroscopy has been performed on complexes **1a** and **1b**, leading to the observation of bands at 1978 cm⁻¹ and 1911 cm⁻¹. (i) These spectra should be shown. (ii) These bands are assigned to the N-N stretching frequencies of the terminal N₂ ligands; i.e., they should correspond to the antisymmetric combination of N-N stretches at the individual Mo units. However, it is to be anticipated that due to the coupling of the two Mo units these bands in turn split into two bands. For the first dinuclear penta(bisdinitrogen) complex of this type, [W(N₂)₂(PEt₂Ph)₃]₂(μ-N₂) such a splitting has in fact been observed (Anderson et al., J. Chem. Soc., Chem. Commun. 1982, 1291). (iii) The symmetric combination of the N-N stretching frequencies of the terminal N₂ ligands should also appear as a small band in the IR spectrum. Moreover, these vibrations as well as the N-N stretch of the bridging ligand should be observable with Raman spectroscopy. (iv) All of these frequencies should be discussed concerning the different ligand moieties". (i) According to the suggestion, the IR spectra of **1a** and **1b** have been added to the Supplementary Information (Supplementary Figures 7 and 8 in the revised Supplementary Information). (ii) As pointed out by Reviewer 2, the splitting is observed for [W(N₂)₂(PEt₂Ph)₃]₂(μ-N₂). All the dinitrogen-bridged dimolybdenum-dinitrogen complexes bearing pincer ligands [Mo(N₂)₂(pincer)]₂(μ-N₂) with similar structures to **1a–1c** have only one strong band assignable to terminal dinitrogen ligands (*Nat. Chem.* **3**, 120 (2011), *JACS* **136**, 9719 (2014), etc.). These experimental results support the present results in the present manuscript. (iii) We have already measured Raman spectroscopy several times. Unfortunately, the spectrum has not been obtained due to the instability. (iv) We have newly added the discussion on the frequencies of the complexes on page 5 in the revised manuscript. Thank you very much for your valuable suggestion.

(9) Reviewer 2 pointed out that "(2) The 15N NMR spectrum of **1b** has been measured. It would be interesting to determine and discuss the coupling constants which are obtained, also in comparison to those of the dinuclear complex **1a**".

As pointed out by Reviewer 2, it is interesting to determine and discuss the coupling constants of **1b** in the ^{15}N NMR. We have newly added the detailed information of ^{15}N NMR coupling constants of **1b** on page 5 in the revised manuscript. However, we have not discussed coupling constants of **1b** in detail in the revised manuscript because of the limitation of space. On the other hand, we have already tried to measure the ^{15}N NMR of **1a**. Unfortunately, we could not obtain the spectrum of **1a** due to its instability under the reaction conditions.

(10) Reviewer 2 pointed out that “(3) Regarding the issue of how the imidazole and benzimidazole based PCP ligands influence the properties of the resulting complexes the papers of Gradert et al. 2013 & 2014 should be cited, where exactly this point has been treated”. According to the suggestions, we have newly added some comments on the influence of imidazole and benzimidazole-based PCP ligands on page 6, including the indicated references. Thank you very much for the valuable comments.

(11) Reviewer 2 pointed out that “(4) There is no obvious reason why the second catalytic experiments (using CrCp^*2 and different proton sources) are performed in a different way than the first experiments. Why do the amounts of metallocene and proton sources are different in both experiments? To get comparable results it would be necessary to use the same equivalents of electron and proton source in both runs”. We have performed the second catalytic experiments in a different way in order to sharpen the difference among the results, and we have added the comment on page 9 in the revised manuscript. Thank you very much for your valuable suggestion.

(12) Reviewer 2 pointed out that “(5) By means of the catalytic experiments of the different benzimidazole based PCP ligands it becomes clear that the methylated ligand is the best catalyst. Can this observation be understood based on spectroscopy (i.e., IR spectroscopy and/or ^{15}N -NMR spectroscopy) coupled to DFT calculations?”. As described in the original manuscript on page 9, we have not yet obtained the exact reason why **1c** worked as the more effective catalyst than **1a**, but we consider that the introduction of two methyl groups to the benzimidazol-2-ylidene skeleton in the PCP-pincer ligand may increase the back-donating ability of molybdenum centers to the coordinated dinitrogen ligands and activate the terminal dinitrogen ligands more strongly than **1a**. The IR spectrum of **1c** in the solid state showed a strong absorption peak assignable to terminal dinitrogen ligands at 1969 cm^{-1} , which is lower than that of **1a** (1978 cm^{-1}). According the suggestion, we have carried out DFT calculations on **1c**

(2012 cm^{-1} for **1a**; 2010 cm^{-1} for **1c**). The result of the DFT calculations supports the experimental results. Thank you very much for your valuable suggestion.

(13) Reviewer 2 pointed out that “(6) Pg. 13: Please document the observation of the formation of the dinitrogen-bridged dimolybdenum complex with loss of one terminal dinitrogen ligand by showing some data”. After a solid **1a** is dried under vacuum for several hours, IR spectrum changed and **1a** was completely converted into paramagnetic compound(s). We have added the IR spectrum of the paramagnetic compound(s) to the Supplementary Figure 10 in the revised version. The results imply that some of the dinitrogen ligands were lost. Thank you for your valuable suggestions.

(14) Reviewer 2 pointed out that “(7) ^{13}C -NMR experiments of some ligands and of the complexes **1a** and **1b** are missing. The measured ^{13}C -NMR spectra should be assigned. In particular, it is important to discuss the shift of the coordinating C-atom of the PCP ligands”. According to the suggestion, the ^{13}C NMR spectra of all the ligands and ligand precursors have been added to the Supplementary Information and all the signals have been assigned. However, we have also tried to measure the ^{13}C NMR spectra of complexes **1a–1c**. Unfortunately, the signals of carbene and some others could not be observed because of their low solubility. Thank you for your valuable suggestions.

As for the comments by Reviewer 3

(15) Reviewer 3 pointed out that “Some issues with prepositions throughout the paper (i.e., at the bottom of page 5: “...or the electron-donating ability of the pincer ligands for the following reasons,” at the bottom of page 10 (and several times throughout the paper): “protonation of a terminal dinitrogen ligand,” to/for mix-ups throughout, etc.)”. According to the suggestion, we have corrected the indicated prepositions. Thank you very much for your careful reading.

(16) Reviewer 3 pointed out that “On page 7 at the end of the first paragraph, the reasoning for why the Mo-N2(equatorial) bond distance in **1b'** is significantly elongated when the dinuclear complex is formed is unclear to me. (“because the symmetrical dimolybdenum structure of **1** specifies the position of the bridging nitrogen at the midpoint of the two molybdenum atoms”...I am not sure why this would necessarily cause the Mo-N2 to

elongate significantly)". To avoid confusion, the corresponding sentences on page 7 in the original manuscript have been deleted in the revised manuscript.

- (17) Reviewer 3 pointed out that "Why is the activity of catalyst 1b the same in the presence of CoCp₂ and without a reductant present at all? Is this just a case of CoCp₂ not being a strong enough reducing agent for this compound?". As pointed out by Reviewer 3, CoCp₂ does not have enough reducing ability for this system.
- (18) Reviewer 3 pointed out that "Along the same lines, what are the redox potentials of 1a and 1b and why is CrCp*₂ so effective? I liked that the authors listed the potentials of the reductants but no conclusions were drawn from them". Since the detailed mechanism of this reaction has not been clarified, the exact reduction potential required for each reaction intermediate is unclear. As a result, we have not yet obtained the reason why CrCp*₂ provided the best result.
- (19) Reviewer 3 pointed out that "I think the term "dinitrogen-bridged dimolybdenum dinitrogen complexes" that is used throughout is unnecessarily wordy. Since you typically are also using a formula or specifying a compound number, you could probably just say "dinitrogen-bridged dimolybdenum complexes" and your readers would figure out that two axial N₂ ligands are also bound". According to the suggestions, we have replaced the term as Reviewer 3 pointed out. Thank you very much for your valuable suggestion.
- (20) Reviewer 3 pointed out that "The differences in Mo-C and N(1)-C(1)-Mo(1)-N5 angle are attributed to steric effects on page 4, but it becomes clear later in the paper that electronic effects are also important here too. I agree that the tBu groups stick out further when the ethylene linkers are used, but the orientation of the NHC ligand is also clearly very important electronically. Certainly Mo-C backbonding is the reason for the shorter Mo-C bond in 1a". According to the suggestions, we have newly added comments "The shortened bond length of Mo(1)-C(1) of **1a** suggests the stronger π -backdonation from the molybdenum center to the NHC due to the almost-perpendicular orientation of the NHC." on page 5 in the revised manuscript. Thank you very much for your valuable suggestion.
- (21) Reviewer 3 pointed out that "Along the same lines, I think that the backbonding between Mo and the NHC ligand as a function of the

orientation of the NHC needs to be brought up earlier in the manuscript. As soon as I saw the crystal structures, this became obvious to me, but it didn't actually get mentioned in the manuscript until the computational section. A reader should be able to understand these logical conclusions without computational data and orbital pictures. You can always refer the reader to the later portion of the paper using "vide infra". I also think the authors should use the very different IR N₂ stretching frequencies to bolster this point. I don't think the interplay between Mo-C backbonding and N₂ stretching frequency was emphasized well in the computational section either – could have been correlated with BDE". According to the suggestions, we have newly added comments "Further information on this topic is discussed based on DFT calculations (*vide infra*)." on page 5 in the revised manuscript. Thank you very much for your valuable comments.

(22) Reviewer 3 pointed out that "Are any of the 3 peaks observed in the IR spectrum of **1b** in solution attributed to remaining N₂-bridged compound?". We have re-examined the IR spectrum of **1b** in THF, where two strong N₂ absorptions were observed. At the same time, we have confirmed that dinuclear species **1b** was almost converted to mononuclear species **1b'** by ³¹P NMR. These results indicate that the two absorptions are assignable to **1b'**. We have newly added the IR spectrum of **1b** in THF as Supplementary Fig. 8 in the revised Supplementary Information. Thank you for your valuable comments.

(23) Reviewer 3 pointed out that "What is the experimental error in the data points listed in Tables 1 and 2? Do each of these represent just one catalytic run? Typically, yields are reported as an average of 3 trials so that experimental error can be taken into account when comparing numbers". According to the suggestion, we have newly added the experimental errors in the data points listed in Tables 1 and 2 in the revised manuscript. Thank you for your valuable comments.

(24) Reviewer 3 pointed out that "The experimental section is quite short and not reflective of all the experimental details in the paper. I think, at the very least, the synthesis of **1b** should be reported here. I realize that much of the experimental detail has been relegated to the Supporting Information file, but I think the reader should be informed of that. For example, a sentence like "Detailed synthetic procedures for ligand precursors and metal precursors **3a-3c** are included in the Supporting Information file." One specific catalytic run was described, but was this procedure also used for all of the catalytic

runs in the paper? If so, this needs to be explicitly stated. If not, the modifications used for each trial need to be described (can be placed in the SI file, as long as the reader is told where to find it)". According to the suggestions, we have newly moved the syntheses of **1b–1c** and **3a–3c** to the "Method" in the revised manuscript from the original Supplementary Information. For the experimental detail of ligand precursors and metal precursors, we have newly added comments as follows: "Detailed synthetic procedures for ligand precursors and metal precursors **3a–3c** are included in the Supplementary Methods." on page 4 in the revised manuscript. For the experimental procedures of catalytic runs, we have added the comments "(for the detailed procedure, see the Supplementary Information)" on pages 9 and 10. Thank you very much for your valuable comments.

(25) Reviewer 3 pointed out that "Compound 1c seemingly comes out of nowhere when the catalysis is being discussed. Is this a new compound or was it previously reported elsewhere? If it is new, it should be included in the "preparation and characterization" section of the manuscript. If it is previously reported, this should be stated explicitly in the manuscript with a reference provided". According the suggestion, we have newly added the synthesis of **1c** to the "preparation and characterization of 1" section in the Results part in the revised manuscript.

(26) Reviewer 3 pointed out that "Similarly, the "General methods" section of the Supporting information files mentions that 4a and 7b were prepared accordingly to literature methods, but these compounds aren't mentioned in the text. In fact, there are lots of new compound numbers defined in the SI file for ligand precursors. I think that a general scheme that shows how all the ligands were synthesized (and defines compound numbers ahead of time) would be useful for the reader before jumping right into synthetic methods". On the ground of format stated by *Nature Communications*, we have not mentioned these compounds in the manuscript. According to the suggestion, we have newly added the general scheme and the definition of the compound number to the Supplementary Figure 11 in the revised Supplementary Information.

(27) Reviewer 3 pointed out that "In the "Accession codes" section of the manuscript, the CCDC numbers for 1c, and 3a-3c are listed. I didn't see these structures mentioned in the paper, and there is no compounds defined as 3c.". According to the suggestion, we have newly defined **3c** at the "Results" section in the revised manuscript and mentioned the structures of **1c** and **3a–3c** in

the Results section (for **1c**) and Method section (for **3a–3c**) in the revised manuscript.

- (28) Reviewer 3 pointed out that “The functional and basis sets used for the DFT calculations should be mentioned in the main text of the paper (even in the "Methods" section would be fine). The reader can then be referred to the SI file for further details”. According to the suggestion, a brief summary of computational methods is given in the “Methods” section in the revised manuscript.
- (29) Reviewer 3 pointed out that “These days, it is customary to include spectral data (1H NMR, 31P NMR) for all new compounds in the Supporting Information file, at least for diamagnetic compounds. This is particularly warranted in this case because proof of bulk purity (elemental analysis, high res mass spec) is not provided for any of the ligands or their precursors”. According to the suggestion, we have newly added the spectral data of all the new compounds to the Supplementary Figures 12–56 in the revised Supplementary Information. Thank you very much for the valuable comment.
- (30) Reviewer 3 pointed out that “Crystallography is generally convincing of the connectivity of the compounds, but there are some errors that need to be addressed or explained prior to publication. 1a- structure is OK. 1b- incompletely modelled solvent disorder. 1c-structure is OK. 3a –appears to be a poor quality structure without all atoms anisotropically refined. 3b - incompletely modelled solvent disorder. 3c – high value of R_{int} needs to be explained”. For the crystallographic analysis of **1b**· $2/3\text{C}_6\text{H}_{14}$, the solvent molecule was heavily disordered thus could not be solved properly. This has been already mentioned in the Supplementary Information. For the crystallographic analysis of **3a**· CH_2Cl_2 , one carbon atom of *tert*-butyl group was slightly disordered, thus was solved isotropically. For this problem, we have newly added the comments to the revised Supplementary Information. For the crystallographic analysis of **3b**· $1/3\text{C}_6\text{H}_{14}$, the solvent molecule was heavily disordered thus could not be solved properly. This has been already mentioned in the Supplementary Information. For the crystallographic analysis of **3c**, the high value of R_{int} is probably due to the metric features of the crystal ($\alpha = \gamma = 90^\circ$, $\beta = 90.3248(15)$), which causes overlapping of many diffraction spots. For this problem, we have newly added the comments “As the beta angle of **3c** is almost 90 degrees, the R_{int} value is not good.” to the revised Supplementary Information.

- (31) Reviewer 3 pointed out that “There are many sections/paragraphs/sentences that could be reworded and/or shortened to be more clear and to the point. (i.e., on page 10: “After the formation of 47 equiv of ammonia...suggesting that the active species derived from 1a were still alive.” This could all be reworded to be more clear and could be significantly condensed, as there is a lot of unnecessary/repetitive info being relayed). I would say "were still active" or "were still present", rather than "were still alive”. According to the suggestions, we have condensed the indicated sentence and replaced the expression as Reviewer 3 pointed out. Thank you very much for your valuable suggestion.
- (32) Reviewer 3 pointed out that “Page 10, end of section – "These results indicate that the catalytic behavior of 1a is quite different from that of 2." I agree with this statement, but think that the authors could use more precise language rather than "catalytic behavior." Perhaps "catalyst activity and stability" would be more specific”. According to the suggestion, we have modified the original expression on page 11 in the revised manuscript. Thank you very much for your valuable suggestion.
- (33) Reviewer 3 pointed out that “Page 12, end of first paragraph states that “we also theoretically confirmed that six- and five coordinate mononuclear dinitrogen complexes, 1a’ and B’, can not be protonated by LutH+.” How was this confirmed? Where is the data for this?”. We tried optimizing a product complex comprised of the protonated **1a’ (1b’)** and Lut, but all initial structures that we tried were relaxed to a reactant complex comprised of **1a’ (1b’)** and LutH⁺, even though the optimization started from the N₂-H⁺ distance of 5 Å. Therefore, we are not able to show any structural data about product complexes and transition states for the protonation step. The same situation was found for [*cis*-Mo(N₂)₂(Bim-PCP[1])] **B’**, and hence **B’** should be attacked by an incoming N₂ or **1a’** to give **1a’** or dimolybdenum complex **1a** before protonation. We have added some sentences on page 12 in the revised manuscript.
- (34) Reviewer 3 pointed out that “Page 12, first two sentences of “Electronic properties of PCP[1] in 1a” section: should the references be 12d instead of 12e? 12d seems to be the correct citation and is also the citation used to cite the same fact on page 9”. According to the suggestion, we have modified the reference number in the revised manuscript. Thank you very much for your careful reading.

- (35) Reviewer 3 pointed out that “Page 12: parameters of the Mo-N₂(terminal) moiety are in Figure 3 (typo)”. According to the suggestion, the indicated typo has been corrected in the revised manuscript. Thank you very much for your careful reading.
- (36) Reviewer 3 pointed out that “Page 14: Why does the coordination of the more electron-donating Bim-PCP[1] ligand to molybdenum increase the Mo-Naxial distances as compared to those of the PNP complex?”. As described on page 15 and Figure 10(b), the π -accepting ability of the NHC decreases the propensity of the Mo atom for π -backdonation to both equatorial and axial N₂ ligands. The π -backdonation of a metal center to coordinated N₂ plays an essential role in the binding and activation of N₂. We have newly added one sentence to describe comparison of the Mo-N₂(axial) bond distances between **1a'** and **2'** on pages 14 and 15 in the revised manuscript.
- (37) Reviewer 3 pointed out that “Page 15: How is the Δq of the dinitrogen ligands defined? It was previously defined as the difference in atomic charge between the mononuclear Mo-N₂ complexes and the free ligands (Bim-PCP[1] and PNP), but this doesn't seem applicable when describing the Δq of the dinitrogen ligands”. In the calculation of Δq , the NPA charges of the Mo atom and three N₂ ligands were set to zero for the free ligands. Thus, the charges assigned to the Mo atom and N₂ ligands in **1a'** and **2'** are identical to the Δq values of them. The gross charges of the Mo(N₂)₃ moiety in **1a'** and **2'** can also be regarded as the amount of electron donated from the pincer ligands. We have modified the corresponding sentences on pages 13 and 14 in the revised manuscript.
- (38) Reviewer 3 pointed out that “Typos in references: ·12e – page # should be 3940. ·13a – remove “No.” According to the suggestion, we have modified the page numbers in the revised manuscript. Thank you very much for your careful reading.

Reviewers' comments:

Reviewer #1 (Remarks to the Author):

In our original review of the manuscript, a number of points were raised concerning the conclusions of the manuscript and the support of these conclusions based on the data presented. While some of these points have been partially address, several important technical details must be resolved prior to publication:

1. Key to the proposed mechanism for catalytic ammonia synthesis is the retention of N₂ on the molybdenum complex even upon dissolving in acid. If true, this is truly remarkable but at the same time, experimental evidence must be presented to support this very unusual behaviour. Solution IR data of the N₂ compound must be obtained; this experiment is essential to support the mechanistic hypothesis.

To address this point, the authors have collected IR spectra of their complexes 1a and 2 upon treatment with 96 equiv. of [LutH][OTf], reporting a color change (clearly indicating that a reaction took place). Unfortunately it was reported that no N₂ bands were observed the respective IR spectra of the compounds due "the overlap with absorptions derived from [LutH]OTf." This critical spectrum must be provided in the revised Supporting Information.

If overlapping bands are complicating, the authors could work up the acidified solutions of 1a and 2 to remove [LutH][OTf] and obtain an IR spectrum of the reaction product free of the overlapping bands? Could the authors have used a smaller excess (or perhaps even a stoichiometric amount) of acid to examine the nature of the reaction product with [LutH][OTf]? If overlapping peaks are still an issue, the acid could be isotopically labeled to shift the bands away from the N₂ stretches, likewise with the Mo complex.

These experiments are critical as catalytic dinitrogen reduction runs begin with exposing the appropriate N₂ complex to excess acid. Therefore, every effort should be made to establish the nature of the catalyst system in acid solution before making conclusions about the mechanism of the dinitrogen reduction.

2. As pointed out in my original review that on the basis of the computational results examining the protonation event in complex I (Figure 8), Path I should be favored over Path II, as the highest barrier step in this mechanism is lower than the highest activation barrier step of Path II. The authors have modified the manuscript to state:

"The calculated results suggest that the PCP system may adopt Path II as a possible reaction pathway for the transformation of dinitrogen into ammonia due to the instability of the six-coordinate diazenide complex A-PCP. However, it is difficult to judge which is a major pathway leading to C-PCP because of the large difference in activation energies for the first step in Paths I and II, although we observed that one of the dinitrogen ligands in 1a was readily dissociated by drying a solid 1a under vacuum (Supplementary Fig. 10)."

While Path II is possible, given that Path I is a competing pathway with a much lower overall activation barrier, Path II becomes heavily disfavored. This point is still not appropriately clarified in the revised manuscript. Additionally, extending the observation that N₂ ligands can dissociate under full vacuum over the course of 18 hours to the mechanism of catalysis seems to be inappropriate; catalytic dinitrogen reduction runs are carried out under 1 atm of N₂.

3. The authors' efforts to show for comparison the IR spectra of complex 1a before and after exposure to vacuum are appreciated. From supplementary figure 10, it is clear that a new band likely assignable to a new molybdenum complex bearing an N₂ ligand appears at 1897 cm⁻¹ in addition to the band corresponding to 1a at 1978 cm⁻¹. Do these results imply that N₂ dissociation is irreversible, or was the sample handled with a rigorous exclusion of N₂ to prevent reversion back to the starting material prior to the collection of the IR spectrum? In either case, experimental details should be added to supplementary figure 10. Could the disappearance of the new band at 1897 cm⁻¹ be observed upon stirring the mixture under an atmosphere of N₂ (ie. reconstituting the starting material)?

Reviewer #2 (Remarks to the Author):

Referee report on the manuscript entitled:

“Remarkable Catalytic Activity of Dinitrogen-Bridged Dimolybdenum-Dinitrogen Complexes Bearing NHC-Based PCP-Pincer Ligands toward Nitrogen Fixation”

By Nishibayashi et al.

The authors have satisfactorily replied to our criticism, with two exceptions:

(1) On the one hand, the authors stress the stability of the complex 1a during catalysis (pg.11 of the manuscript), on the other hand they state that, due to the instability of 1a, no ^{15}N data could be obtained on this complex. This seems somewhat contradictory to us.

(2) No experimental evidence is provided for the contention that the reactive cycle starts with loss of N_2 (cf Fig.8, Path II).

To support their claim the authors argue that upon drying complex 1a in vacuum it loses one dinitrogen ligand. However, it is not possible to compare the behavior of the bridged molybdenum complexes in bulk material and in solution. Thus the hypothesis that dinitrogen ligands dissociate upon drying under vacuum (20 h!, see Figure 10, Supp Inf) does not prove that the conversion of N_2 into NH_3 using one of the catalysts starts with the loss of N_2 .

To actually show this further experimental evidence would be necessary; e.g. an IR spectrum of the reaction solution. On the other hand, the ^{31}P NMR spectrum of complex 1a in solution is 100% clean, indicating that the complex is stable in solution. So we believe that the authors' hypothesis is not supported by the available experimental data.

Furthermore, the IR spectrum of complex 1a obtained after drying under vacuum only shows one(!) additional band as compared to the pristine material, at 1897 cm^{-1} . The original IR band at 1978 cm^{-1} remains at its position. One interpretation could be that one part of the dimer stays intact (showing its original N-N stretch) whereas the other part loses one N_2 ligand, giving rise to the new band at 1897 cm^{-1} . However, the dimer would lose its inversion symmetry in this case, and a third stretch from the bridging N_2 should become IR allowed. An alternative explanation of the vibrational data would imply no loss of N_2 at all. In the PNP system, the N-N stretching frequency of the bridging dinitrogen ligand was determined as 1890 cm^{-1} . Couldn't the new band at 1897 cm^{-1} also derive from the bridging N_2 ligand? Maybe this vibration gets IR-allowed by a distortion of the dimer in the solid.

Minor points:

(1) Figure 1 and 2 are shown in the manuscript as well as in the Supporting Information. Please delete the one in the Supporting Information.

(2) Page 5: (i) The NMR spectrum shown in Figure 2 is not the ^{15}N -NMR spectrum of 1b; it is the ^{15}N -NMR spectrum of ^{15}N -1b.

(3) Denotation of dinitrogen-bridged dimolybdenum complexes: $[\{\text{Mo}(\text{N}_2)_2(\text{PCP})\}_2(\mu\text{-N}_2)]$

(4) Page 16: The backdonation [...] weakens

(5) NMR spectrum of 7a: In the experimental data CDCl_3 was used as the NMR solvent. In the caption of Figure 33-35 (Supp Inf) it is written that THF-d_8 was used as the solvent. Which is correct?

(6) Page 5 (line 3): NHC unit

Reviewer #3 (Remarks to the Author):

The authors have adequately addressed most of my concerns and those of the other referees, although I am still not satisfied with the level of rigor with which the crystallographic aspects of this manuscript have been approached. Disordered solvent that can't be modelled is typically removed using PLATON SQUEEZE. I am not qualified to assess the effect of the unit cell parameters (the beta angle) on R_{int} , so I can't judge whether this explanation is sufficient. I urge the editors to consult a crystallographic referee prior to acceptance. Otherwise, this paper is now appropriate for publication in Nature Communications.

Reviewer #4 (Remarks to the Author):

Crystallographic Report.

The structures reported should be fit for the authors' purposes - but several are essentially not finished and thus the paper should be accepted only after these have been completed to a reasonable standard.

1b and 3b. Both contain areas of electron density that the authors describe as "disordered solvent" and which they model by putting 1/2 C atoms on the largest Q peaks. This is not acceptable. The authors must either find a way of constructing a sensible disorder model or, failing that, they should use the PLATON/SQUEEZE methodology to deal with the problem. Without access to the *.res files and hkl files it is hard to be sure, but a crystallographic specialist should be able to do this easily. I note that the problems for these two structures are not limited to the solvent. The displacement ellipsoids for the complex also indicate problems, to such an extent that I would not put any weight onto details such as accurate geometric parameters. A last point is that 1b is in the unusual Ccca space group and Rint is large. The authors should investigate the possibility that this is a twinned sample with true lower symmetry. If this is not so then they should comment on their investigations in the ESI or cif.

3a. Such a poor quality model should be flagged as such in the main text. It is not obvious what the main problem here is. I note that there are very high Q peaks that have not been assigned to atoms - more unmodelled solvent perhaps? The authors need to attempt to model such large features. Or at least they have to report (ESI) in detail where these features are, how many of them there are and why they cannot be modelled. A butyl group has been left with an isotropic C atom. This is not acceptable. From the neighbouring displacement ellipsoids a disorder model may be required here.

3c. This has a high Rint, $Z = 2$ and a monoclinic beta angle of approx 90 degrees. There is a high chance that this is a twin. The authors should investigate this and either re-refine as a twin or report in the ESI/cif why it is that they think it is not a twin.

As for the comments by Reviewer 1

(1) Reviewer 1 pointed out that “1. Key to the proposed mechanism for catalytic ammonia synthesis is the retention of N₂ on the molybdenum complex even upon dissolving in acid. If true, this is truly remarkable but at the same time, experimental evidence must be presented to support this very unusual behaviour. Solution IR data of the N₂ compound must be obtained; this experiment is essential to support the mechanistic hypothesis. To address this point, the authors have collected IR spectra of their complexes **1a** and **2** upon treatment with 96 equiv. of [LutH][OTf], reporting a color change (clearly indicating that a reaction took place). Unfortunately, it was reported that no N₂ bands were observed the respective IR spectra of the compounds due “the overlap with absorptions derived from [LutH]OTf.” This critical spectrum must be provided in the revised Supporting Information. If overlapping bands are complicating, the authors could work up the acidified solutions of **1a** and **2** to remove [LutH][OTf] and obtain an IR spectrum of the reaction product free of the overlapping bands? Could the authors have used a smaller excess (or perhaps even a stoichiometric amount) of acid to examine the nature of the reaction product with [LutH][OTf]? If overlapping peaks are still an issue, the acid could be isotopically labeled to shift the bands away from the N₂ stretches, likewise with the Mo complex. These experiments are critical as catalytic dinitrogen reduction runs begin with exposing the appropriate N₂ complex to excess acid. Therefore, every effort should be made to establish the nature of the catalyst system in acid solution before making conclusions about the mechanism of the dinitrogen reduction.”

According to the suggestion, we have newly measured the IR spectrum in the stoichiometric reaction of **2** with 2 equiv of [LutH]OTf in toluene at room temperature for 2 min under an atmospheric pressure of dinitrogen. This result indicates that the formation of a dinitrogen-bridged dimolybdenum–hydrazide complex [Mo(OTf)(≡NNH₂)(PNP)–N≡N–Mo(N₂)₂(PNP)]OTf due to two absorptions at 1804 cm⁻¹ and 1984 cm⁻¹, which are assigned as bridging dinitrogen ligand and terminal dinitrogen ligand, respectively. However, these peaks disappeared in the stoichiometric reaction of **2** with 5 equiv of [LutH]OTf under the same reaction conditions although a new peak appeared at 2001 cm⁻¹. We believe that this experimental result supports our proposal that the first protonation of the terminal dinitrogen ligand in **2** with 1 equiv of [LutH]OTf may occur with maintenance of the dinitrogen-bridged dimolybdenum core. As described in our previous paper (H. Tanaka, et al., *Nat. Commun.* **5**, 3737 (2014)), we observed the formation of both dinuclear nitride complex bearing the dinitrogen-bridged dimolybdenum core [Mo(OTf)(PNP)(≡N)–N≡N–Mo(N₂)(PNP)] and mononuclear nitride complex [Mo(OTf)(≡N)(PNP)] by mass spectrometry from the stoichiometric reaction of **2** with 2 equiv of [LutH]OTf at room temperature in toluene. Furthermore, we observed

the formation of dinuclear ammonia complex bearing the dinitrogen-bridged dimolybdenum core $[\text{Mo}(\text{NH}_3)(\text{PNP})-\text{N}\equiv\text{N}-\text{Mo}(\text{N}_2)_2(\text{PNP})]$ by mass spectrometry from the catalytic reaction of **2** with excess amounts of CoCp_2 and $[\text{LutH}]\text{OTf}$ under the same reaction conditions.

In contrast to the reactivity of **2**, we did not observe the formation of a dinitrogen-bridged dimolybdenum-hydrazide complex in the stoichiometric reaction of **1a** with 2 equiv of $[\text{LutH}]\text{OTf}$ under the same reaction conditions although one absorption appeared at 1972 cm^{-1} in IR spectrum. However, we observed the formation of dinuclear hydrazide complex bearing the dinitrogen-bridged dimolybdenum core $[\text{Mo}(\text{OTf})(\text{PCP})(\text{NNH}_2)-\text{N}\equiv\text{N}-\text{Mo}(\text{PCP})]$ by mass spectrometry from the stoichiometric reaction of **1a** with 2 equiv of $[\text{LutH}]\text{OTf}$. Furthermore, we observed the formation of both dinuclear nitride complex bearing the dinitrogen-bridged dimolybdenum core $[\text{Mo}(\text{OTf})(\text{PCP})(\equiv\text{N})-\text{N}\equiv\text{N}-\text{Mo}(\text{N}_2)_2(\text{PCP})]$ and mononuclear nitride complex $[\text{Mo}(\text{OTf})(\equiv\text{N})(\text{PCP})]$ by mass spectrometry from the catalytic reaction of **1a** with excess amounts of CrCp^*_2 and $[\text{LutH}]\text{OTf}$ under the same reaction conditions. Unfortunately, we cannot conclude that we have obtained direct evidence to support our proposal that the first protonation of the terminal dinitrogen ligand in **1a** with 1 equiv of $[\text{LutH}]\text{OTf}$ may occur with maintenance of the dinitrogen-bridged dimolybdenum core. As described in the present manuscript, the dissociation energy of the bridging dinitrogen ligand from molybdenum atom in the dinitrogen-bridged dimolybdenum complex is substantially different between **1a** and **2**. We believe that this is one of the reasons why the reactivity of **1a** is different from that of **2**.

As described in the present manuscript, in sharp contrast to **1a** and **2**, **1b** has no catalytic activity toward the catalytic nitrogen fixation under ambient reaction conditions although the terminal dinitrogen ligands in **1b** are more activated than those in **1a**. This is due to the instability of the dinuclear core in **1b** in solution. In fact, as shown in Fig. 2a and Fig. 2b in the re-revised manuscript and in the previous paper (K. Arashiba et al. *Nat. Chem.* **3**,120 (2011)), ^{15}N NMR spectra of **1a**- $^{15}\text{N}_2$, **1b**- $^{15}\text{N}_2$, and **2**- $^{15}\text{N}_2$ showed that **1a** and **2** maintain their dinuclear structure in solution and **1b** does not. These experimental results indicate that the maintenance of the dinitrogen-bridged dimolybdenum core such as **1a** and **2** is necessary to promote the first protonation of the terminal dinitrogen ligand. We strongly ask this reviewer to remind our proposal that the first protonation of the terminal dinitrogen ligand in **1a** and **2** with 1 equiv of $[\text{LutH}]\text{OTf}$ may occur with maintenance of the dinitrogen-bridged dimolybdenum core. As proposed in our previous paper (H. Tanaka, et al., *Nat. Commun.* **5**, 3737 (2014)), the dimolybdenum core does not have to be maintained at all reaction steps in the catalytic cycle.

Based on these experimental results, as to the first protonation of the terminal dinitrogen ligand in **1a**, we have proposed a similar reaction pathway with that we proposed in the previous paper (H. Tanaka, et al., *Nat. Commun.* **5**, 3737 (2014)), where **2**

worked as a catalyst. However, at the present reaction system by using **1a** as a catalyst, we cannot completely exclude other possibility that mononuclear molybdenum complexes bearing a PCP-pincer ligand worked as key reactive intermediates. Taking into the consideration pointed out by this reviewer, we have newly added some comments on the proposal reaction pathway for **1a** that we cannot completely exclude other possibility that mononuclear molybdenum complexes bearing a PCP-pincer ligand worked as key reactive intermediates in the re-revised manuscript. We have also added the sentence that we consider that further study is necessary before we conclude the reaction pathway for **1a** in the re-revised manuscript. According to the suggestion, we have newly added IR spectra in stoichiometric reactions of **1a** and **2** as Supplementary Fig. 7–12, in the re-revised Supplementary Information. Thank you very much for the valuable suggestions.

(2) Reviewer 1 pointed out that “2. As pointed out in my original review that on the basis of the computational results examining the protonation event in complex I (Figure 8), Path I should be favored over Path II, as the highest barrier step in this mechanism is lower than the highest activation barrier step of Path II. The authors have modified the manuscript to state: “The calculated results suggest that the PCP system may adopt Path II as a possible reaction pathway for the transformation of dinitrogen into ammonia due to the instability of the six-coordinate diazenide complex A-PCP. However, it is difficult to judge which is a major pathway leading to C-PCP because of the large difference in activation energies for the first step in Paths I and II, although we observed that one of the dinitrogen ligands in **1a** was readily dissociated by drying a solid **1a** under vacuum (Supplementary Fig. 10).” While Path II is possible, given that Path I is a competing pathway with a much lower overall activation barrier, Path II becomes heavily disfavored. This point is still not appropriately clarified in the revised manuscript. Additionally, extending the observation that N₂ ligands can dissociate under full vacuum over the course of 18 hours to the mechanism of catalysis seems to be inappropriate; catalytic dinitrogen reduction runs are carried out under 1 atm of N₂.”

As pointed out by Reviewer 1, we have reinvestigated the reaction pathway shown in Figure 8 of the revised manuscript. As a result, we consider that **Path II** in Figure 8 is not likely to be a major pathway for the first protonation process in the PCP system due to the significantly high activation barrier compared to the barrier for the protonation in **Path I**. According to the suggestion, we have deleted the discussion about **Path II** in the main text of the re-revised manuscript. We really appreciate your valuable comments again.

(3) Reviewer 1 pointed out that “The authors’ efforts to show for comparison the IR spectra of complex **1a** before and after exposure to vacuum are appreciated. From

supplementary figure 10, it is clear that a new band likely assignable to a new molybdenum complex bearing an N₂ ligand appears at 1897 cm⁻¹ in addition to the band corresponding to **1a** at 1978 cm⁻¹. Do these results imply that N₂ dissociation is irreversible, or was the sample handled with a rigorous exclusion of N₂ to prevent reversion back to the starting material prior to the collection of the IR spectrum? In either case, experimental details should be added to supplementary figure 10. Could the disappearance of the new band at 1897 cm⁻¹ be observed upon stirring the mixture under an atmosphere of N₂ (ie. reconstituting the starting material)?”.

As pointed out by Reviewer 1, the IR spectra of **1a** before and after exposure to vacuum are not enough to support the proposed reaction pathway such as **Path II** shown in Figure 8 of the revised manuscript. As described in the comments (2) by Reviewer 1, taking into the consideration by Reviewer 1, we have deleted the discussion about **Path II** in Figure 8 in the main text of the re-revised manuscript. Because we no longer need to claim that the catalytic cycle begins with the loss of one of the terminal dinitrogen ligands (**Path II** in Figure 8) in the re-revised manuscript, we have deleted the experimental result that one of the dinitrogen ligand in **1a** was readily dissociated by drying a solid **1a** under vacuum in the main text of the re-revised manuscript. We have also deleted Supplementary Fig 10 of the revised Supplementary Information in the re-revised Supplementary Information. Thank you very much for your valuable comments.

As for the comments by Reviewer 2

- (4) Reviewer 2 pointed out that “On the one hand, the authors stress the stability of the complex **1a** during catalysis (pg.11 of the manuscript), on the other hand they state that, due to the instability of **1a**, no ¹⁵N data could be obtained on this complex. This seems somewhat contradictory to us.”

As pointed out by Reviewer 2, we have newly performed ¹⁵N₂ labeling experiments of **1a** with improved procedures. As a result, we have succeeded in measuring a ¹⁵N NMR spectrum of **1a** in a benzene (C₆D₆) solution. We have newly added the spectrum to Figure 2 (a) in the re-revised manuscript. Thank you very much for your suggestion.

- (5) Reviewer 2 pointed out that “No experimental evidence is provided for the contention that the reactive cycle starts with loss of N₂ (cf Fig.8, Path II). To support their claim the authors argue that upon drying complex **1a** in vacuum it loses one dinitrogen ligand. However, it is not possible to compare the behavior of the bridged molybdenum complexes in bulk material and in solution. Thus the hypothesis that dinitrogen ligands dissociate upon during drying under vacuum (20 h!, see Figure 10, Supp Inf) does not prove that the conversion of N₂ into NH₃ using one of the catalysts starts with the loss of N₂. To actually show this further

experimental evidence would be necessary; e.g. an IR spectrum of the reaction solution. On the other hand, the ^{31}P NMR spectrum of complex **1a** in solution is 100% clean, indicating that the complex is stable in solution. So we believe that the authors' hypothesis is not supported by the available experimental data. Furthermore, the IR spectrum of complex **1a** obtained after drying under vacuum only shows one(!) additional band as compared to the pristine material, at 1897 cm^{-1} . The original IR band at 1978 cm^{-1} remains at its position. One interpretation could be that one part of the dimer stays intact (showing its original N-N stretch) whereas the other part loses one N_2 ligand, giving rise to the new band at 1897 cm^{-1} . However, the dimer would lose its inversion symmetry in this case, and a third stretch from the bridging N_2 should become IR allowed. An alternative explanation of the vibrational data would imply no loss of N_2 at all. In the PNP system, the N-N stretching frequency of the bridging dinitrogen ligand was determined as 1890 cm^{-1} . Couldn't the new band at 1897 cm^{-1} also derive from the bridging N_2 ligand? Maybe this vibration gets IR-allowed by a distortion of the dimer in the solid.”

As pointed out by Reviewer 2, the IR spectra of **1a** before and after exposure to vacuum are not enough to support the proposed reaction pathway such as **Path II** shown in Figure 8 of the revised manuscript. As described in the comments (2) by Reviewer 1, taking into the consideration by Reviewers 1 and 2, we have deleted the discussion about **Path II** in Figure 8 in the main text of the re-revised manuscript. Because we no longer need claim that the catalytic cycle begins with the loss of one of the terminal dinitrogen ligands (**Path II** in Figure 8) in the re-revised manuscript, we have deleted the experimental result that one of the dinitrogen ligand in **1a** was readily dissociated by drying a solid **1a** under vacuum in the main text of the re-revised manuscript. We have also deleted Supplementary Fig 10 of the revised Supplementary Information in the re-revised Supplementary Information. Thank you very much for your valuable comments.

- (6) Reviewer 2 pointed out that “Minor points 1. Figure 1 and 2 are shown in the manuscript as well as in the Supporting Information. Please delete the one in the Supporting Information.”

According to the suggestions, we have deleted the Supplementary Figures 1 and 2 of the revised Supplementary Information in the re-revised Supplementary Information. Thank you very much for your careful reading.

- (7) Reviewer 2 pointed out that “Minor points 2. Page 5: (i) The NMR spectrum shown in Figure 2 is not the ^{15}N -NMR spectrum of **1b**; it is the ^{15}N -NMR spectrum of ^{15}N -**1b**.”

According to the suggestion, we have corrected the point indicated by Reviewer 2 in the re-revised manuscript. Thank you very much for your careful reading.

(8) Reviewer 2 pointed out that “Minor points 3. Denotation of dinitrogen-bridged dimolybdenum complexes: $[\{\text{Mo}(\text{N}_2)_2(\text{PCP})\}_2(\mu\text{-N}_2)]$ ”.

According to the suggestion, we have corrected the points indicated by Reviewer 2 in the re-revised manuscript. Thank you very much for your careful reading.

(9) Reviewer 2 pointed out that “Minor points 4. Page 16: The backdonation [...] weakens”.

Thank you for your careful reading. However, as described in the comments (2) by Reviewer 1, according to the suggestion by Reviewer 1, we have deleted the sentence in the re-revised manuscript.

(10) Reviewer 2 pointed out that “Minor points 5. NMR spectrum of **7a**: In the experimental data CDCl_3 was used as the NMR solvent. In the caption of Figure 33-35 (Supp Inf) it is written that $\text{THF-}d_8$ was used as the solvent. Which is correct?”.

Although Reviewer 2 pointed out that CDCl_3 was used as a solvent for NMR spectra of **7a**, we used $\text{THF-}d_8$ as a solvent for NMR spectra of **7a** as shown in Supplementary Figures 33–35 of the revised Supplementary Information. Thank you very much for your careful reading.

(11) Reviewer 2 pointed out that “Minor points 6. Page 5 (line 3): NHC unit”. According to the suggestion, we have corrected the points indicated by Reviewer 2 in the re-revised manuscript. Thank you very much for your careful reading.

As for the comments by Reviewer 3

(12) Reviewer 3 pointed out that “Disordered solvent that can't be modelled is typically removed using PLATON SQUEEZE”.

According to the suggestion, we have corrected the models for disordered solvents of **1b**· $1/3\text{C}_6\text{H}_{14}$ and **3b**· $1/3\text{C}_6\text{H}_{14}$ in the re-revised Supplementary information.

For the crystal of **1b**· $1/3\text{C}_6\text{H}_{14}$ (0.75 molar **1b** and 0.25 molar hexane per asymmetric unit), we have reconstructed a new model of hexane consisting of four located carbon atoms (C(36) to C(39)) with atom occupancies of 0.375 (6/16), where one hexane molecule (six carbon atoms) is disordered among four asymmetric units to form a 16-membered ring (C(36)–C(37)–C(38)–C(39)–C(39)*–C(38)*–C(37)*–C(36)*–C(36)'–C(37)'–C(38)'–C(39)'–C(39)''–C(38)''–C(37)''–C(36)''). Positions of hydrogen atoms of hexane cannot be refined because of this disorder. Thus, we have not further applied PLATON/SQUEEZE methodology. As a result, we have revised the crystallographic report in the re-revised Supplementary information.

For the crystal of **3b**·1/3C₆H₁₄, we have reanalyzed another new crystal and have reconstructed a new model of hexane consisting of five located carbon atoms (C(24) to C(28)) solved as a rigid group with atom occupancies of 0.4 (6/15), where one hexane molecule (six carbon atoms) is disordered among three asymmetric units to form a 15-membered ring (C(24)–C(25)–C(26)–C(27)–C(28)–C(24)*–C(25)*–C(26)*–C(27)*–C(28)*–C(24)'–C(25)'–C(26)'–C(27)'–C(28)'). Positions of hydrogen atoms of hexane cannot be refined because of this disorder. Thus, we have not further applied PLATON/SQUEEZE methodology. As a result, we have revised the crystallographic report in the re-revised Supplementary information. Thank you very much for your valuable comments.

As for the comments by Reviewer 4

(13) Reviewer 4 pointed out that “**1b** and **3b**. Both contain areas of electron density that the authors describe as "disordered solvent" and which they model by putting 1/2 C atoms on the largest Q peaks. This is not acceptable. The authors must either find a way of constructing a sensible disorder model or, failing that, they should use the PLATON/SQUEEZE methodology to deal with the problem. Without access to the *.res files and hkl files it is hard to be sure, but a crystallographic specialist should be able to do this easily. I note that the problems for these two structures are not limited to the solvent. The displacement ellipsoids for the complex also indicate problems, to such an extent that I would not put any weight onto details such as accurate geometric parameters. A last point is that **1b** is in the unusual *Ccca* space group and R_{int} is large. The authors should investigate the possibility that this is a twinned sample with true lower symmetry. If this is not so then they should comment on their investigations in the ESI or cif.”

According to the suggestion, we have corrected the models for disordered solvents of **1b**·1/3C₆H₁₄ and **3b**·1/3C₆H₁₄ in the re-revised Supplementary information.

For the crystals of **1b**·1/3C₆H₁₄, we have investigated the raw diffraction data by using R-AXIS RAPID AUTO Ver. 3.11, which has ruled out the possibility of both non-merohedral and merohedral twins. Thus, choice of *Ccca* space group is appropriate for the crystal of **1b**·1/3C₆H₁₄. By the revision of the crystallographic data with revised model of disordered hexane without applying the PLATON/SQUEEZE methodology as mentioned above, the R_{int} value has decreased from 0.1011 to 0.0778, and displacement ellipsoids have been slightly improved. The maximum Q peak (2.18) is located at the position of (0.122, 0.250, 0.249) almost the same with that of the Mo(2) atom, ruling out the existence of any other solvents. As a result, we have revised the crystallographic report in the re-revised Supplementary information.

For the crystals of **3b**·1/3C₆H₁₄, we have reanalyzed another new crystal and have revised the crystallographic data with revised model of disordered hexane without applying the PLATON/SQUEEZE methodology as mentioned above. The maximum Q

peak (1.66) is located at the position of (0.292, 0.539, 0.326) 0.53 Å apart from the C(27) atom, but other Q peaks are small enough to rule out the existence of any other solvents. As a result, we have revised the crystallographic report in the re-revised Supplementary information.

Thank you very much for your valuable comments.

(14) Reviewer 4 pointed out that “**3a**. Such a poor quality model should be flagged as such in the main text. It is not obvious what the main problem here is. I note that there are very high Q peaks that have not been assigned to atoms - more unmodelled solvent perhaps? The authors need to attempt to model such large features. Or at least they have to report (ESI) in detail where these features are, how many of them there are and why they cannot be modelled. A butyl group has been left with an isotropic C atom. This is not acceptable. From the neighbouring displacement ellipsoids a disorder model may be required here.”

To avoid unnecessary discussion on the X-ray analysis, we have deleted the X-ray crystallographic data of **3a** in the re-revised Supplementary information. Thank you very much for your careful reading.

(15) Reviewer 4 pointed out that “**3c**. This has a high R_{int} , $Z' = 2$ and a monoclinic beta angle of approx 90 degrees. There is a high chance that this is a twin. The authors should investigate this and either re-refine as a twin or report in the ESI/cif why it is that they think it is not a twin.”

According to the suggestion, we have investigated the raw diffraction data of **3c**·0.5CH₂Cl₂ by using R-AXIS RAPID AUTO Ver. 3.11, which has ruled out the possibility of both non-merohedral and merohedral twins. We have revised the crystallographic data, where the R_{int} value has decreased slightly from 0.1429 to 0.1268. We have revised the crystallographic report in the re-revised Supplementary information. Thank you very much for your careful reading.

Reviewers' comments:

Reviewer #1 (Remarks to the Author):

In our original review of the first submission by Nishibayashi and coworkers, my summary was as follows:

“The manuscript can be generally considered as having two major parts. The first section details the synthesis, characterization and catalytic activity of the new molybdenum compounds while the second part of the manuscript serves to rationalize the observed reactivity trends with mechanistic studies and computational (DFT) analysis of frontier orbitals and bonding.”

The review was concluded with the note:

“In summary, this work is interesting and could potentially shed light on some valuable design principles for synthesizing more effective catalysts for N₂ reduction. However, the authors must revisit and clarify their arguments and rationalization for the observed high catalytic activity of the newly synthesized PCP complexes.”

Unfortunately, the authors have altogether deleted the rationalization of the high catalytic activity of their complexes. It is clear that the original two rounds of submissions were flawed and the origin of the high catalytic performance of the new complex is not understood. I note that “high” represents a 4-fold increase of what was known in the literature previously.

This is summed up the response of Nishibayashi and coworkers in the cover letter accompanying the manuscript:

“[...] we no longer need to claim that the catalytic cycle begins with the loss of one of the terminal dinitrogen ligands (Path II in Figure 8) in the re-revised manuscript[.]”

While this certainly eliminates the “problem areas” of the report, it also significantly detracts from the impact of the manuscript and now no design principles for the synthesis of high-activity PCP Mo dinitrogen fixation catalysts. This point is exacerbated by the fact that despite the omission of a key component of the manuscript, the authors still claim in the abstract that:

“We believe that our findings described in this article provide an opportunity to design more effective nitrogen fixation system.”

However, no experimental evidence is presented in the manuscript that is sufficient to draw rigorous conclusions about the high activity of the newly designed PCP Mo complexes.

I have reviewed this manuscript now three times with the understanding that an alternative and/or revised explanation will be presented for the high catalytic activity of the presented PCP molybdenum systems that is of interest to the broad readership of Nature Communications. It is my opinion that the novelty of this manuscript lies in a deeper

understanding of the systems presented in the form of well-founded structure-activity relationships and the elucidation of key design principles that aid in the future designs of previous systems without that it is an incremental advance using protocols now described many times by many research groups.

A good benchmark for gauging progress on this front is provided by the authors themselves in the introduction to the current manuscript:

“During our continuous study, we have realized three promising clues to develop more effective catalysts. The first clue is the introduction of an electron-donating group to the pincer ligands to increase the backdonating ability of the molybdenum atom to the dinitrogen ligand.⁴² In fact, dinitrogen-bridged dimolybdenum complexes bearing the electron-donating group-substituted PNP-pincer ligands worked as more effective catalysts in our previous reaction system. The second clue is the inhibition of the dissociation of the pincer ligand from the molybdenum atom to increase the stability of the molybdenum complex. ⁴¹ We generally observed the dissociation of the PNP-pincer ligand after the ammonia formation in the catalytic reaction. The third clue is the preservation of the dinitrogen-bridged dimolybdenum core to promote the catalytic ammonia formation from the coordinated dinitrogen.⁴⁴”

In light of these revisions, and in the absence of broad design principles that guide the observations high catalytic activity of PCP Mo complexes, publication of this manuscript in a journal intended for a specialized audience seems more appropriate.

Reviewer #2 (Remarks to the Author):

The authors have satisfactorily replied to our criticism, with the following two exceptions:

(1) ^{15}N NMR data: Please discuss the observed broad singlet of the $\text{N}\alpha$ atom. Actually this nitrogen atom should generate a dt signal due to the coupling to the phosphorus atoms and the $\text{N}\beta$ atom.

(2) Section "Reaction pathway catalyzed by 1a":

The authors write in the new paragraph on pg.12:

"On the reactivity of mononuclear Mo– N_2 complexes with LutH^+ , we confirmed that $[\text{Mo}(\text{N}_2)_3(\text{Bim-PCP}[1])]$ 1a' and $[\text{Mo}(\text{N}_2)_3(\text{Im-PCP}[2])]$ 1b' cannot be protonated by LutH^+ in a similar manner as $[\text{Mo}(\text{N}_2)_3(\text{PNP})]$.⁴⁴ All attempts to optimize a product complex comprised of the protonated 1a' (1b') and Lut resulted in formation of a reactant complex comprised of 1a' (1b') and LutH^+ , even though the optimization started from a structure with the $\text{N}_2 \cdots \text{H}^+$ distance of 5 Å. Thus, the dinuclear Mo– $\text{N}\equiv\text{N}$ –Mo structure is essential to the first protonation step of the PCP complexes, what is more, for the PCP system to exhibit the catalytic activity."

This conclusion, however, does not explain the activity increase of the PCP systems as compared to the PNP system (which relies on the dinuclear structure as well).

In this context, one has to infer from Figure 8 that replacement of the PNP by a PCP ligand does not change the reaction energies and activation energies involved in the first protonation step. The conclusion from this Figure is therefore that the significant increase in catalytic activity of the PCP systems as compared to the PNP system is NOT due to the energetics of the first protonation reaction. The activity increase of the PCP system thus must be due to one of the subsequent steps of the mechanistic cycle.

The results of the calculations should be discussed on the basis of this result.

Reviewer #4 (Remarks to the Author):

Crystallography.

The authors have dealt with most of my original points.

However, I still have problems with their treatment of the solvent in 1b.

What they have modelled cannot be correct as the geometry makes no sense (the bonds angles are much too linear for an alkane chain).

My recommendation for this structure stands as before. Please supply a sensible model for disordered hexane or use SQUEEZE to remove this effect and report such in a suitable manner.

As for the comments by Reviewer 1

(1) Reviewer 1 pointed out that “In our original review of the first submission by Nishibayashi and coworkers, my summary was as follows: “The manuscript can be generally considered as having two major parts. The first section details the synthesis, characterization and catalytic activity of the new molybdenum compounds while the second part of the manuscript serves to rationalize the observed reactivity trends with mechanistic studies and computational (DFT) analysis of frontier orbitals and bonding.” The review was concluded with the note: “In summary, this work is interesting and could potentially shed light on some valuable design principles for synthesizing more effective catalysts for N₂ reduction. However, the authors must revisit and clarify their arguments and rationalization for the observed high catalytic activity of the newly synthesized PCP complexes.” Unfortunately, the authors have altogether deleted the rationalization of the high catalytic activity of their complexes. It is clear that the original two rounds of submissions were flawed and the origin of the high catalytic performance of the new complex is not understood. I note that “high” represents a 4-fold increase of what was known in the literature previously. This is summed up the response of Nishibayashi and coworkers in the cover letter accompanying the manuscript: “[...] we no longer need to claim that the catalytic cycle begins with the loss of one of the terminal dinitrogen ligands (Path II in Figure 8) in the re-revised manuscript [...]” While this certainly eliminates the “problem areas” of the report, it also significantly detracts from the impact of the manuscript and now no design principles for the synthesis of high-activity PCP Mo dinitrogen fixation catalysts. This point is exacerbated by the fact that despite the omission of a key component of the manuscript, the authors still claim in the abstract that: “We believe that our findings described in this article provide an opportunity to design more effective nitrogen fixation system.” However, no experimental evidence is presented in the manuscript that is sufficient to draw rigorous conclusions about the high activity of the newly designed PCP Mo complexes. I have reviewed this manuscript now three times with the understanding that an alternative and/or revised explanation will be presented for the high catalytic activity of the presented PCP molybdenum systems that is of interest to the broad readership of Nature Communications. It is my opinion that the novelty of this manuscript lies in a deeper understanding of the systems presented in the form of well-founded structure-activity relationships and the elucidation of key design principles that aid in the future designs of previous systems without that it is an incremental advance using protocols now described many times by many research groups. A good benchmark for gauging progress on this front is provided by the authors themselves in the introduction to the current manuscript: “During our continuous

study, we have realized three promising clues to develop more effective catalysts. The first clue is the introduction of an electron-donating group to the pincer ligands to increase the backdonating ability of the molybdenum atom to the dinitrogen ligand.⁴² In fact, dinitrogen-bridged dimolybdenum complexes bearing the electron-donating group-substituted PNP-pincer ligands worked as more effective catalysts in our previous reaction system. The second clue is the inhibition of the dissociation of the pincer ligand from the molybdenum atom to increase the stability of the molybdenum complex.⁴¹ We generally observed the dissociation of the PNP-pincer ligand after the ammonia formation in the catalytic reaction. The third clue is the preservation of the dinitrogen-bridged dimolybdenum core to promote the catalytic ammonia formation from the coordinated dinitrogen.⁴⁴ In light of these revisions, and in the absence of broad design principles that guide the observations high catalytic activity of PCP Mo complexes, publication of this manuscript in a journal intended for a specialized audience seems more appropriate.” According to the suggestion pointed out by Reviewer 1, we deleted **Path II** as another reaction pathway shown in Figure 8 of the re-revised manuscript. As a result, Reviewer 1 pointed out the absence of broad design principles that guides the high catalytic activity of PCP Mo complexes. We do not agree with Reviewer 1. Since Path II is energetically unfavorable when compared to Path I as Reviewer 1 pointed out in the previous comments, the deletion of Path II in the re-revised manuscript is not “deletion of the rationalization of the high catalytic activity of their complexes” nor “detraction from the impact of the manuscript and now no design principles for the synthesis of high-activity PCP Mo dinitrogen fixation catalysts”. As pointed out by Reviewer 1, no significant difference in the reaction pathway and energy profiles was observed between Mo-PCP and Mo-PNP complexes shown in Figure 8 of the re-revised manuscript. However, we believe that no difference is interesting to consider the role of the PCP ligand in the Mo complex. Thus, the result shown in Figure 8 of the re-revised manuscript suggests that the introduction of PCP ligand instead of PNP ligand to the Mo complex does not dramatically change the energy profiles but substantially affect the stability towards decomposition of the Mo complex. Taking the comments pointed out by Reviewer 1 into consideration, we have modified the final part of results of the re-revised manuscript accordingly. In fact, as described in the revised version of the re-revised manuscript, we have concluded that the stability of the Mo complex bearing a strong bonding ability of the ligand to the Mo atom is more important than the activation energy in the reaction pathway and energy profiles. According to the revision, we have changed the order of Figures 8–10 in the revised version of the re-revised manuscript. As a result, we believe that the present paper has still a broad interest to be published in *Nature Communications*. Thank you very much for your valuable comments.

As for the comments by Reviewer 2

- (2) Reviewer 2 pointed out that “¹⁵N NMR data: Please discuss the observed broad singlet of the N α atom. Actually this nitrogen atom should generate a dt signal due to the coupling to the phosphorus atoms and the N β atom.” As pointed out by Reviewer 2, we observed broad singlet of the N α atom shown in Figure 2 (a) of the re-revised manuscript. Unfortunately, we have not yet clarified the exact reason, however, a similar broad singlet was previously observed in the same type of W–dinitrogen complex reported by other research group (*J. Chem. Soc., Dalton Trans.* 1986, 245). As a result, we have newly added some comments and the corresponding reference in the revised version of the re-revised manuscript. Thank you very much for your suggestion.
- (3) Reviewer 2 pointed out that “Reaction pathway catalyzed by 1a”: The authors write in the new paragraph on pg.12: “On the reactivity of mononuclear Mo–N₂ complexes with LutH⁺, we confirmed that [Mo(N₂)₃(Bim-PCP[1])] 1a' and [Mo(N₂)₃(Im-PCP[2])] 1b' cannot be protonated by LutH⁺ in a similar manner as [Mo(N₂)₃(PNP)].⁴⁴ All attempts to optimize a product complex comprised of the protonated 1a' (1b') and Lut resulted in formation of a reactant complex comprised of 1a' (1b') and LutH⁺, even though the optimization started from a structure with the N₂ ... H⁺ distance of 5 Å. Thus, the dinuclear Mo–N≡N–Mo structure is essential to the first protonation step of the PCP complexes, what is more, for the PCP system to exhibit the catalytic activity.” This conclusion, however, does not explain the activity increase of the PCP systems as compared to the PNP system (which relies on the dinuclear structure as well).” We guess that this comment is due to the misunderstandings of Reviewer 2. What we described here is that the reactivity of mononuclear Mo-PCP complexes toward the protonation suggests the importance of the dinuclear Mo-NN-Mo structure, *in a similar manner as* the PNP system.
- (4) Reviewer 2 pointed out that “In this context, one has to infer from Figure 8 that replacement of the PNP by a PCP ligand does not change the reaction energies and activation energies involved in the first protonation step. The conclusion from this Figure is therefore that the significant increase in catalytic activity of the PCP systems as compared to the PNP system is NOT due to the energetics of the first protonation reaction. The activity increase of the PCP system thus must be due to one of the subsequent steps of the mechanistic cycle. The results of the calculations should be discussed on the basis of this result.” As described in the manuscript, the first protonation is expected to be the most unfavorable process in the catalytic cycle. As pointed out by Reviewer 2, no significant difference in the reaction pathway and energy profiles was observed between Mo-PCP and Mo-PNP complexes shown in Figure 8 of the re-revised manuscript. At present, we have not yet observed any different behaviors between Mo-PCP and Mo-PNP complexes in the subsequent steps of the catalytic cycle. However, we believe that no difference is interesting to consider

the role of the PCP ligand in the Mo complex. Thus, the result shown in Figure 8 of the re-revised manuscript suggests that the introduction of PCP ligand instead of PNP ligand to the Mo complex does not change the energy profiles of the reaction but affect the stability toward decomposition of the catalyst as shown in Figure 7 in the re-revised manuscript. Taking the comments pointed out by Reviewer 2 into consideration, we have modified the final part of results of the re-revised manuscript accordingly. In fact, as described in the revised version of the re-revised manuscript, we have concluded that the stability of the Mo complex bearing a strong bonding ability of the ligand to the Mo atom is more important than the activation energy in the reaction pathway and energy profiles. According to the revision, we have changed the order of Figures 8–10 in the revised version of the re-revised manuscript. Thank you very much for your valuable comments.

As for the comments by Reviewer 4

(5) Reviewer 4 pointed out that “I still have problems with their treatment of the solvent in **1b**. What they have modelled cannot be correct as the geometry makes no sense (the bonds angles are much too linear for an alkane chain). My recommendation for this structure stands as before. Please supply a sensible model for disordered hexane or use SQUEEZE to remove this effect and report such in a suitable manner.” According to the suggestion, we have used SQUEEZE to remove the solvent molecules in the crystal of **1b**. We have corrected the crystallographic data for **1b** in the revised version of the re-revised Supplementary information. Thank you very much for your valuable comments.